# Optimization of fluorinated phenyl azides as universal photocrosslinkers for semiconducting polymers

Zhao-Siu Tan[1,3], Zaini Jamal[1,3], Desmond W. Y. Teo[1], Hor-Cheng Ko[1], Zong-Long Seah[2], Hao-Yu Phua[2], Peter K. H. Ho [2], Rui-Qi Png[2] ✉ & Lay-Lay Chua [1] ✉

Fluorinated phenyl azides (FPA) enable photo-structuring of π-conjugated polymer films for electronic device applications. Despite their potential, FPAs have faced limitations regarding their crosslinking efficiency, and more importantly, their impact on critical semiconductor properties, such as charge-carrier mobility. Here, we report that azide photolysis and photo-crosslinking can achieve unity quantum efficiencies for specific FPAs. This suggests preferential nitrene insertion into unactivated C–H bonds over benzazirine and ketenimine reactions, which we attribute to rapid interconversion between the initially formed hot states. Furthermore, we establish a structure–activity relationship for carrier mobility quenching. The binding affinity of FPA crosslinker to polymer π-stacks governs its propensity for mobility quenching in both PM6 and PBDB-T used as model conjugated polymers. This binding affinity can be suppressed by FPA ring substitution, but varies in a non-trivial way with π-stack order. Utilizing the optimal FPA, photocrosslinking enables the fabrication of morphology-stabilized, acceptor-infiltrated donor polymer networks (that is, PBDB-T: ITIC and PM6: Y6) for solar cells. Our findings demonstrate the exceptional potential of the FPA photochemistry and offer a promising approach to address the challenges of modelling realistic molecular interactions in complex polymer morphologies, moving beyond the limitations of Flory–Huggins mean field theory.

Since the discovery more than a decade ago that FPA photo-crosslinking can produce high performance vertical and lateral heterostructures for semiconducting polymer devices[1], including morphology-stabilized donor–acceptor networks for polymer organic solar cells[2], numerous clever device architectures exploiting FPA photocrosslinking have emerged, for example, for light-emitting diodes based on polymers[3,4], quantum dots[5], and perovskites[6], electrochromic devices[7], organic field-effect transistors[8–10], organic solar cells[11], and organic photodiodes[12,13]. Crosslinking has also enabled unconventional material compositions, such as charge-doped layers with immobilized molecular dopants[14]. Alternative crosslinking moieties include diazirines[15,16], alkyl azide[17], vinyl[18,19], and epoxy[20].

FPAs and diazirines are the two most important families of C–H insertion crosslinkers[21,22]. For these two families, photon absorption causes cleavage of the azide or diazirine moiety to nitrogen gas, and a singlet nitrene or carbene, respectively, that can insert into unactivated C–H bonds. This crosslinking mechanism advantageously makes use of the C–H bonds that are copiously present in polymers as

[1]Department of Chemistry, National University of Singapore, Lower Kent Ridge Road, S117552 Singapore, Singapore. [2]Department of Physics, National University of Singapore, Lower Kent Ridge Road, S117550 Singapore, Singapore. [3]These authors contributed equally: Zhao-Siu Tan, Zaini Jamal. ✉e-mail: phypngrq@nus.edu.sg; chmcll@nus.edu.sg

crosslinking sites. Since it does not require two specific chemical functionalities to react together, crosslinking can be accomplished potentially at the theoretical minimum concentration of the crosslinker. The photospeed is limited by the crosslinker photolysis quantum yield $\varphi_P$, that is, the probability for cleavage to occur upon photon absorption, while the crosslinking efficiency $\xi_{XL,P}$ is the fraction of cleaved crosslinkers that provides effective crosslinks, measured in our work as the inverse of the ratio of actual to theoretical crosslinker concentration to reach gel point of the polymer matrix, where an infinite network emerges.

However, nitrenes have long suffered from a reputation for poor efficiency compared to carbenes for insertion into C–H bonds[23–25]. Numerous parasitic side reactions are known. For example, the singlet nitrene ($^1$N) from phenyl azide isomerizes quickly to benzazirine (Bza) and cyclic ketenimine (Kti, i.e. 1,2,4,6-azacycloheptatetraene) or undergoes intersystem crossing to the triplet nitrene ($^3$N)[26]. Thus, phenyl azides fail to give significant insertion yields in liquid media, unlike phenyl diazirines[27]. Although perfluorinated phenyl azides give much higher insertion yields[28,29], which enable FPAs to be used as photoaffinity labels[26,30], the reputation did not go away. *Ortho, ortho*-difluorination (relative to azide position) does retard the parasitic Bza and Kti pathways[31], but the insertion yields remain modest at room temperature, $\lesssim 50\%$ for liquid toluene as model C–H substrate, rising to $\geq 80\%$ only when frozen at $-196\,°C$[32,33]. Thus, the literature is replete with assertions that azides are inferior to diazirines for photocrosslinking.

A critical question remains: what are the values of $\varphi_P$ and $\xi_{XL,P}$ for FPA photocrosslinkers in practical polymer matrices. Prior studies have shown that $\xi_{XL,P}$ can reach unity for certain bisFPAs in polystyrene (PS)[1]. This question is of particular significance for semiconducting polymer films, which may be degraded by excessive light exposure and deleterious by-products. The other critical question is the nature of the carrier traps, whether chemical or morphological, that are generated in photocrosslinked films, and how these could be suppressed. Furthermore, a lower $\xi_{XL,P}$ would necessitate a higher crosslinker concentration which would exacerbate the situation.

In this Report, we investigated a family of substituted FPA photocrosslinkers and showed that azide photolysis proceeds generally with unity $\varphi_P$. Notably, numerous members also achieve perfect crosslinking efficiency with unity $\xi_{XL,P}$ in PS, signifying fully suppressed Bza and Kti pathways in solid polymer matrices. This is further supported by FTIR photoreaction analysis. Using PM6 and PBDB-T as practical conjugated polymer models, we further observed superefficient crosslinking, where $\xi_{XL,P}$ even surpasses unity, that is, the gel point occurring well below the theoretical value for a given polymer molecular weight. This is likely due to an increase in the effective molecular weight of the polymer through π-stacking. We further demonstrate that the carrier traps are morphological in origin, arising from π-interactions between the crosslinker and the polymer backbone. This is evidenced by a structure–activity relationship that links the decrease in carrier mobility to the binding affinity of the FPA moiety to the polymer backbone. This binding affinity can be modulated by substituents on the FPA ring, but the affinity sequence varies with π-stack order of the polymer. The findings here refine the previously proposed heuristic design rule based on "steric substitution"[1], and provide crucial molecular design insights for engineering polymer–molecule interactions. Finally, we demonstrate using the optimal FPA photocrosslinker that high-efficiency morphology-stabilized, acceptor-infiltrated donor-polymer networks can be fabricated for PM6: Y6 and PBDB-T: ITIC solar cells.

## Results and discussion
### Thermophysical properties
The set of FPA photocrosslinkers studied comprises FPA0 and FPA1 which have unsubstituted 4-azido-2,3,5,6-tetrafluorophenyl moieties, and carbonyl or carbonyloxyethyleneoxycarbonyl bridge, respectively,

between the FPA rings (Fig. 1a); FPA8a and 8b, which have single and double *meta*-methyl (Me) substitution, respectively, relative to azide; and FPA6a and 6b, single and double *meta*-isopropyl (*i*-Pr) substitution, respectively; and finally, 2FPA1, a tetrakisazide linked by the tetraesterified pentaerythritol, which has been widely employed in the literature[9,11,34,35], and which may be regarded as a dimerized FPA1. This series facilitates the systematic study of substituent and polyfunctionalization effects, inspired by the previous success of FPA6a.

Thermogravimetry reveals a weight loss step between 150 and 200 °C due to azide thermolysis (Fig. 1b), while differential scanning calorimetry reveals a sharp endothermic peak due to melting and a broad exothermic peak due to azide decomposition (Fig. 1c). The onset loss temperature where the combined rate of decomposition and sublimation (if any) first exceeds 1% min$^{-1}$, is $125 \pm 5\,°C$ (Supplementary Table 1). Thus, polymer films formulated with FPAs can tolerate short processing bakes at up to 120 °C, which is sufficient to dry off residual solvents. The azide decomposition is exothermic. But the Yoshida shock sensitivity and explosive propagation indices are both near zero or marginally negative, which suggests these FPAs are inherently safe (Supplementary Table 1). The decomposition temperature and enthalpy are sufficiently high and low, respectively, to avoid explosive behavior.

### Electronic transitions
Ultraviolet spectroscopy reveals a strong absorption at the 254-nm deep-ultraviolet (DUV) line emitted by low-pressure Hg vapor lamps (Fig. 1d). FPAs are thus suitable as DUV photocrosslinkers. FPA0 shows a redshift of its absorption band, due to π-conjugation of its two FPA moieties, while 2FPA1 shows the expected doubling of molar absorptivity compared with FPA1. The peak wavelength and molar absorptivity are generally well reproduced by time-dependent DFT/CAM-B3LYP/6-31 G calculations in SMD with acetonitrile as solvent (Fig. 1e). Thus, the electronic transitions contributing to the DUV band are well understood.

This band arises from transition to the excited state ES4 for FPA0, and the pair of degenerate excited states ES3 and ES4 for the others. Please see the molecular model and selected molecular orbital diagrams for the one-electron excitations in Fig. 1f, g, respectively. These are the lowest-lying optically-coupled excited states. They comprise π–π* excitations primarily from (HOMO and HOMO−1) to (LUMO and LUMO + 1). In contrast, the lowest excited state ES1 is a dark state which lies 0.7–0.9 eV below (ES3, ES4), comprising one-electron excitation to a LUMO+$n$ that has strong π*-antibonding character in the azide group (Fig. 1h). These features correspond to those well known in phenyl azides[36,37].

### Photolysis quantum efficiency
Azide photolysis is a first-order reaction whose time constant $t_o$ can be determined from the mono-exponential decay of the surviving azide fraction: $\frac{n}{n_o} = \exp(-\frac{t}{t_o})$, where $t$ is the exposure time (Fig. 2a). We performed these measurements in PS: FPA films, where the monodispersed 200-kD PS contains 5–10 w/w % (weight/weight) of the FPA. $t_o$ was then converted to $\varphi_P$, based on photon irradiance, photo-absorption cross-section, and the absorptance enhancement factor due to optical interference of the transmitted and reflected DUV light (Fig. 2b). The latter was evaluated following the usual transfer matrix method, as interference determines the actual fraction of photons absorbed[38,39].

The $\varphi_P$ values are found to be unity for all FPAs within experimental uncertainty (Supplementary Table 2). Every absorbed photon leads to the cleavage of an azide group. The perfect quantum yield likely arises from the strong π*-antibonding azide character in ES1 and the large reaction exogenicity. Thus, FPAs have the highest possible photospeed for their absorption cross-section. This is vastly superior to diazirines, for example, 3-trifluoromethyl-3-phenyldiazirine, where up to one-third may photo-isomerize to the relatively stable diazo

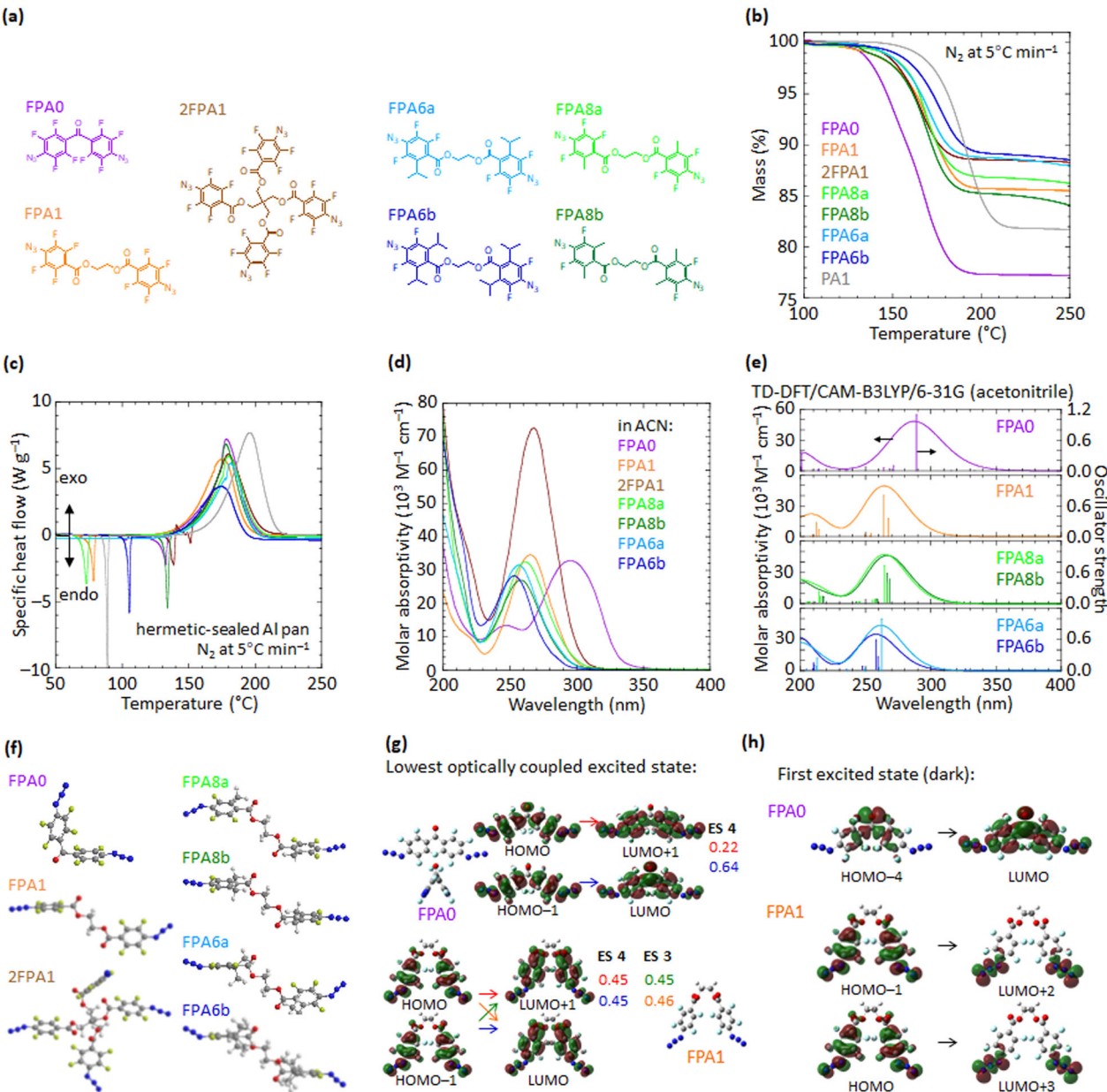

**Fig. 1 | Basic characterization of the FPA photocrosslinkers. a** Chemical structure. **b** Thermogravimetry characteristics. **c** Differential scanning calorimetry characteristics, color coded same as (**b**). **d** Solution-state absorption spectra; **e** Computed absorption spectra for vertical excitations at the optimized ground-state geometry ($S_o$). **f** Molecular model: blue, nitrogen; yellow, fluorine; red, oxygen; gray, carbon; white, hydrogen. **g** Molecular orbital wavefunctions for dominant one-electron excitation contributions to ES4 of FPA0, and to both ES3 and ES4 of FPA1, with corresponding coefficients given. **h** Dominant one-electron excitation contributions to ES1 of FPA0 and FPA1. The wavefunctions for FPA1 are also similar to those of other substituted FPAs in this work. PA1 is the non-fluorinated analogue of FPA1.

isomer[22,27]. The perfect quantum yield here also contrasts with the modest yield for phenyl azides in solution ($\varphi_P < 0.6$)[40,41].

## Crosslinking efficiency in polystyrene model

The $\xi_{XL,P}$ values are also found to be unity in the PS matrix for several members of the FPA family: FPA1, 2FPA1, FPA8a, and FPA6a (Supplementary Table 2). These are evaluated as the inverse of the ratio of the actual crosslinker concentration $C_c$ for the gel point to its theoretical value $C_{c,th}$ or $C'_{c,th}$ (Fig. 2c). The retention characteristic empirically follows $\zeta = 1 - (C_c/C)^n$, where $n$ is a fitted characteristic exponent, typically 1.5–2[1]. $\xi_{XL,P}$ depresses from unity with the second alkyl substituent, more so for di-$i$-Pr than di-Me substitution, and also for FPA0. Unity $\xi_{XL,P}$ implies that the C–H insertion yield is unity. Every crosslinker makes an effective crosslink without participating in unproductive side reactions. In marked contrast, photolysis of per-fluorophenyl azides in liquid media, such as toluene or cyclohexane, typically gives insertion yields lower than 50%, which corresponds to $\xi_{XL}$ values lower than 0.25 in liquid media[30,32,33].

The perfect crosslinking efficiency implies the usual Bza, Kti and ³N pathways are fully suppressed within the PS matrix. Thus, FPAs provide clean photocrosslinking. Thus, they are superior to alkyl azides and diazirines[15,22,25,42].

## Photoreaction product analysis

In situ photoreaction product FTIR analysis confirms the occurrence of the desired photochemistry, and the identity of the side reactions where

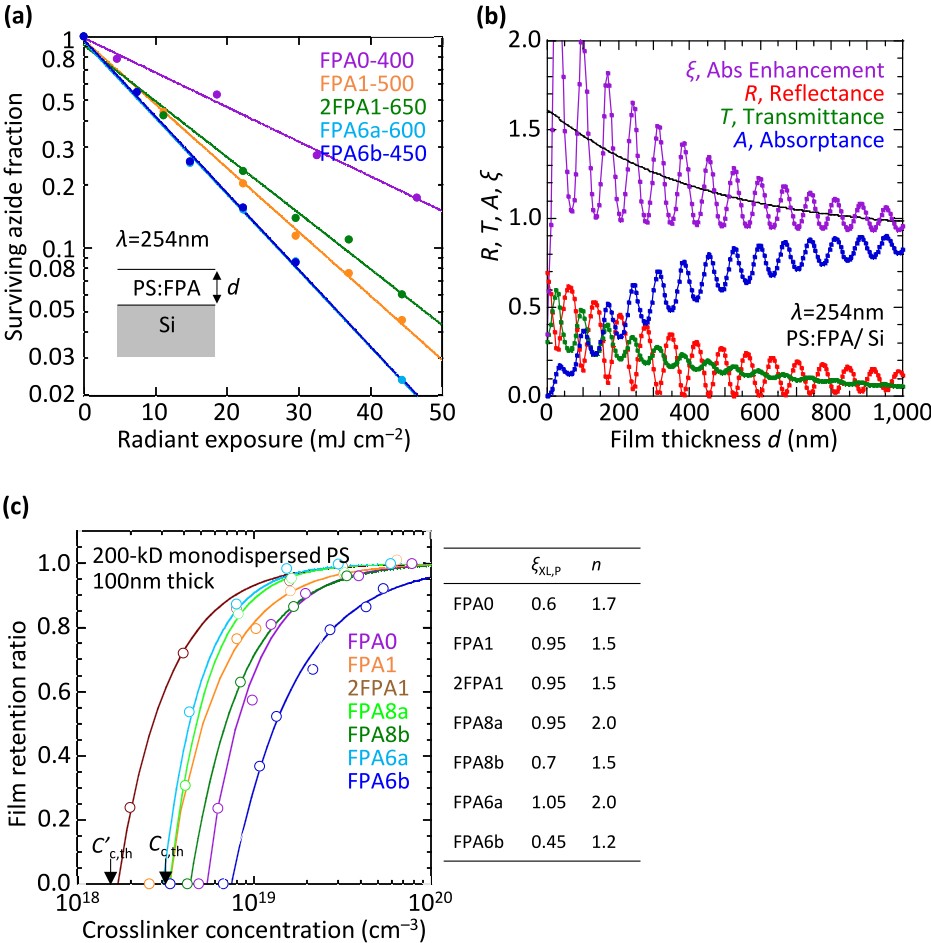

**Fig. 2 | Photolysis characteristics in polystyrene: FPA films. a** Plot of surviving azide fraction against radiant exposure at 254-nm wavelength, measured by FTIR spectrometry of the asymmetric azide stretching mode ($v_{as}$ N$_3$, *ca.* 2125 cm$^{-1}$), for selected FPAs in PS: PFA films with thickness *d* given (in nm) after sample name. **b** Plot of reflectance, transmittance, and absorptance, together with absorptance enhancement factor given by: $A/(1-\exp(-\alpha d))$, where $\alpha$ is inverse absorption length ($4\pi \kappa/\lambda = 2.0 \times 10^4$ cm$^{-1}$), computed for parallel-beam illumination. The relevant optical constants at 254-nm-wavelength are: Si, 1.60 – 3.75*i*; and PS: FPA, 1.79 – 0.04*i*. Interference oscillations are damped by the elongated tube lamp illumination and film thickness variation, where the damped function (fitted black curve)

follows: $A + B*\exp(-\beta d)$, where $A$ is $(1-R)$ at large film thicknesses (0.9196), $B$ is 0.695, and $\beta$ is $2.4 \times 10^4$ cm$^{-1}$. **c** Film retention characteristics for 200-kD polystyrene. Lines are fit to $\zeta = 1 - (C_c/C)^n$, where $\zeta$ is the ratio of film thickness after solvent wash to original film thickness, $C$ is crosslinker concentration, $C_c$ is the experimental gel point, and $n$ is the characteristic exponent. The theoretical gel points for this PS are marked by $C_{c,th}$ and $C'_{c,th}$ at 3.2 and $1.6 \times 10^{18}$ cm$^{-3}$ for bisazide and tetrakisazide, respectively. A number concentration of $1.0 \times 10^{19}$ cm$^{-3}$ corresponds to a weight concentration of about 0.77 w/w% for the FPAs here. Table gives $\xi_{XL,PS}$ and $n$ values, with estimated uncertainties (1 sd) of 0.05 and 0.15, respectively.

$\xi_{XL,P} < 1$. To attain the required sensitivity and baseline drift of less than $10^{-3}$ absorbance unit per hour, the spectrometer was operated in an ambient-stabilized cleanroom[43]. The difference spectra after photolysis reveal the expected loss of azide (N$_3$: $v_{as}$ 2125 cm$^{-1}$, $v_s$ 1233–1286 cm$^{-1}$; Fig. 3) and gain of –NH– and alkyl/aryl crosslinks (NH: $v$ 3395 cm$^{-1}$; C–N: $v$ 1200–1330 cm$^{-1}$) for all PS: FPAs, and additional gain of Kti features[44,45] for PS: FPA8b and PS: FPA6b (C=N=C[46,47]: $v$ 1830 cm$^{-1}$; hydrolysis product CONH[48]: $v$ 3200 cm$^{-1}$; rearrangement product C≡N[49] : $v$ 2285 cm$^{-1}$). Thus, the Bza and Kti pathways that are widely observed in liquid media are not, in fact, inherent to FPA photolysis in solid matrices, but revived by a second Me or *i*-Pr substitution on the FPA ring. Furthermore, the FPA-derived amine (FPA$n$-NH$_2$), a signature product of $^3$N[50], occurs less than 5% (NH$_2$: $v_{as}$ 3496 cm$^{-1}$, $v_s$ 3392 cm$^{-1}$). This is lower than the 10–30% in liquid media[30,32,33]. Finally, the loss of $\xi_{XL,P}$ for FPA0 is not due to such side reactions. Thus, it must be attributed to the formation of intrachain crosslinks due to the short length of the crosslinker.

## Reaction profile
The computed reaction profiles for the anticipated isomerization of the photogenerated singlet nitrene for all FPAs suggest a sizeable

activation energy barrier that does not decrease for FPA8b and 6b (Fig. 4a). Thus, the interconversions could not have taken place on the adiabatic surface as widely presumed in the literature. We evaluated the transformations amongst the ground states of$^1$:N ⇄ Bza ⇄ Kti, using unrestricted Kohn-Sham DFT/CAM-B3LYP/6-31 G(*d*), extending the seminal work of Borden and Platz, and their co-workers[51,52]. These calculations give similar results as those of CASSCF(8,8)/6-31 G(*d*) for fluorinated phenylnitrene $^1$N ⇄ Bza, albeit 0.35 eV higher[51]. The results suggest the closed-shell singlet nitrene (denoted $^1$N*)–formed by N$_2$ ejection from the azide ES1 ($^1$N$_3$*)[37,53,54]–lies *ca.* 0.75 eV above the ground-state open-shell $^1$N, which in turn lies *ca.* 0.65 eV above $^3$N. Increasing alkyl ring substitution raises the relative energy of Bza, and particularly Kti, which raises the activation energies for $^1$N → Bza and Bza → Kti. The first activation energy is sizeable (0.85–0.95 eV), and, as expected, much larger than for the nonfluorinated azide[51].

To reconcile these results with the facile observation of Kti products in liquid media and also for FPA8b and 6b, we postulate here that the interconversions occur amongst vibrationally hot states in solid matrices, not the ground states$^1$:N (hot) ⇄ Bza (hot) ⇄ Kti (hot) (Fig. 4b). The $^1$N$_3$* lies considerably (≳3 eV) above the $^1$N + N$_2$. The excess energy

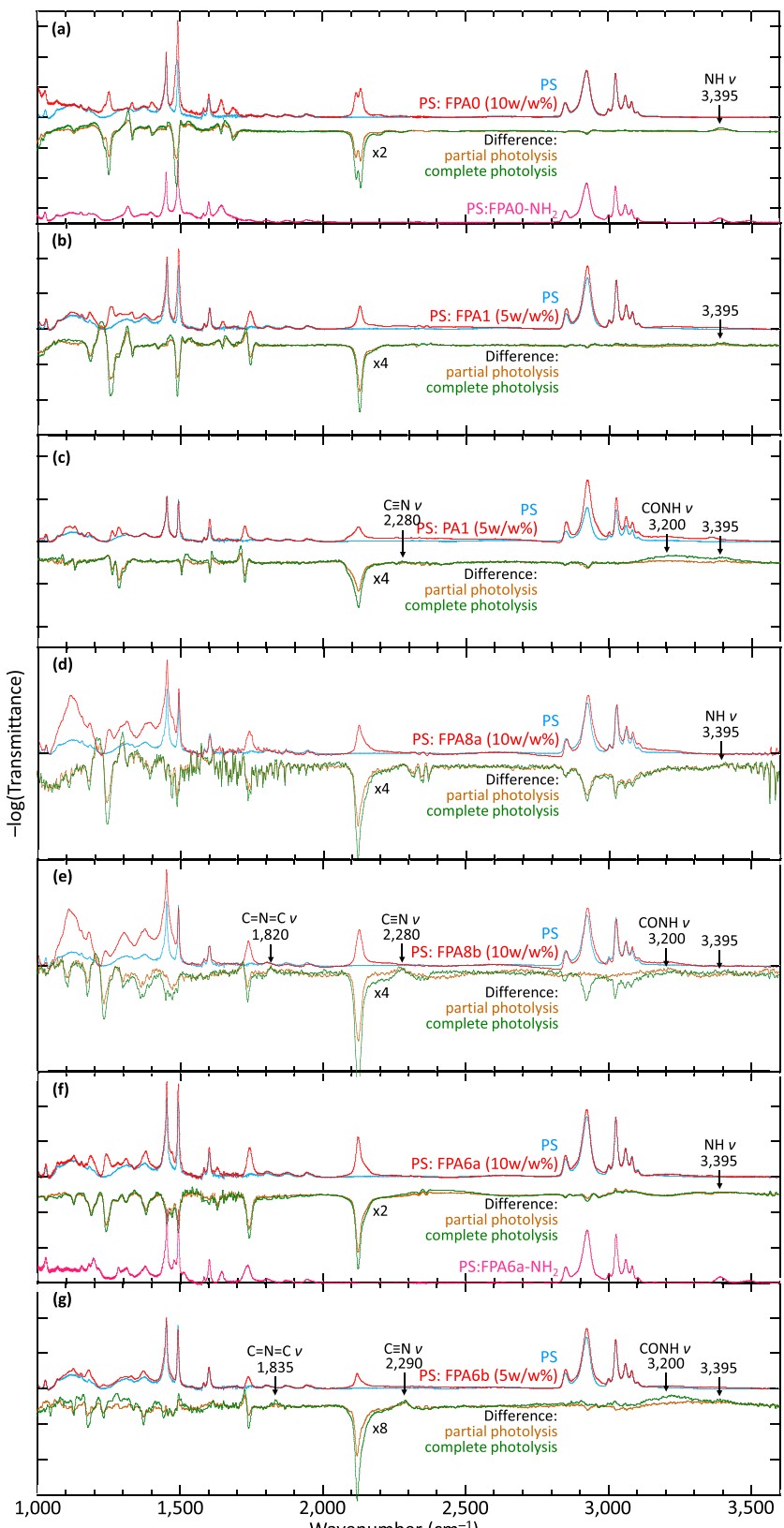

**Fig. 3 | Photoreaction product analysis in thin films.** FTIR spectroscopy for PS: FPA films before, and after partial or complete photolysis, together with selected authentic amino derivatives as reference. **a** FPA0, **b** FPA1, **c** PA1, **d** FPA8a, **e** FPA8b, **f** FPA6a, and **g** FPA6b. Vertical division, typically 25 milli-absorbance units (mAU); baseline correction error typically less than 1 mAU. The broad "bump" feature at 1120 cm⁻¹ is contributed by substrate $SiO_2$, while the sharp features at 2349 cm⁻¹ are due to imperfect correction for atmospheric $CO_2$, and those at 1400–1900 cm⁻¹ are due to atmospheric $H_2O$. The difference spectra were obtained for the same film, thus giving changes in the crosslinker and PS matrix, independent of the substrate. PA1 is the non-fluorinated analogue of FPA1.

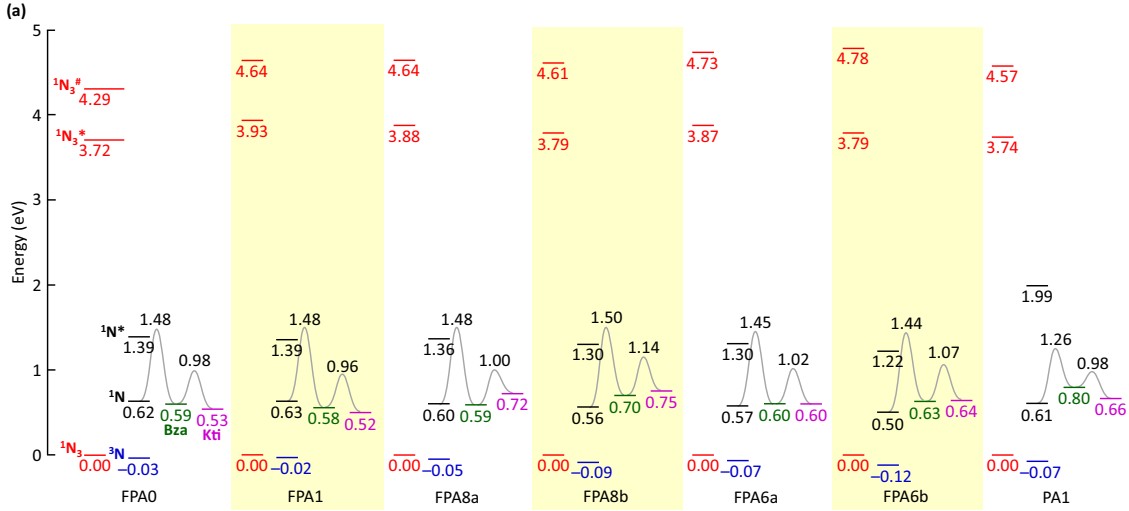

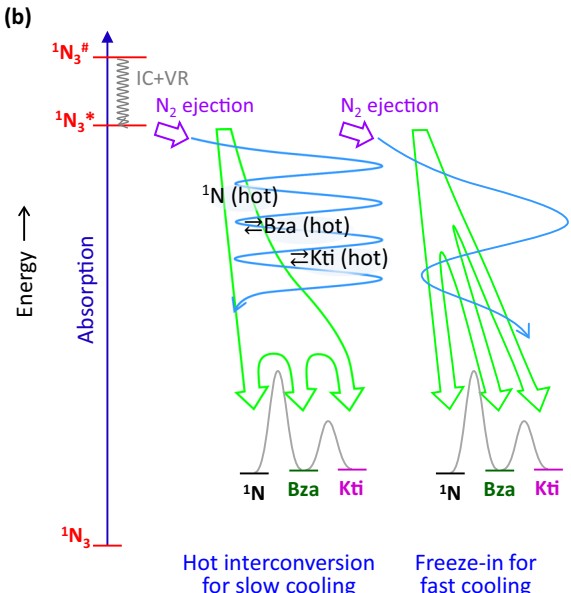

**Fig. 4 | Computed internal energies. a** Reaction profiles for FPAs.$^1$N$_3$ is azide ground state[1], N$_3$* is lowest excited singlet azide (dark)[1], N$_3$# is lowest optically-coupled excited singlet azide[1], N is open-shell singlet nitrene ground state[1], N* is lowest closed-shell singlet nitrene[3], N is ground-state triplet nitrene, Bza is benzazirine, Kti is cyclic ketenimine. Transition state energies for $^1$N → Bza → Kti are shown. For FPA8a and FPA6a, transition states are shown for the lower-energy product where the N atom bridges and then inserts at the C atom away from Me and *i*-Pr, respectively. The other product is a few tens of meV higher in energy. Energies are computed with unrestricted Kohn-Sham DFT/CAM-B3LYP/6-31 G(*d*). Levels represent zero-point internal energies at 298 K relative to $^1$N$_3$, combining the internal energies of the products, i.e., nitrene + N$_2$, after photolysis.$^1$N$_3$* and $^1$N$_3$# are vertical energies from TD-DFT at the same level of theory in acetonitrile. **b** Proposed hot interconversions: $^1$N ⇄ Bza ⇄ Kti, for slow cooling in solid matrices; and freeze-in for fast cooling in liquid media or in the presence of a high density of C–H bonds on the FPA moiety. IC is internal conversion, and VR is vibrational relaxation.

may not efficiently dissipate in solid matrices, unlike in liquid media, where vibrational to translational and rotational mode conversions can occur. Since $^1$N provides the sole escape channel (giving C–H insertion), $\xi_{XL,P}$ can approach unity in solid matrices. But increasing the density of C–H bonds on the FPA moiety adds high-frequency vibrational modes (*ca.* 2900 cm$^{-1}$) that appear to facilitate faster cooling, which traps the interconverting Bza and Kti states.

This postulate is also consistent with available literature data. Evidence for the predicted prompt generation of Kti can be adduced from pump–probe spectroscopy of the fluorinated phenyl azide. The broad absorption feature over 350–400-nm wavelength that emerges within 1 ns was thought to be $^1$N[31,52], which is now ruled out by calculations (Supplementary Fig. 1), but in fact, characteristic of Kti[32]. This

feature already emerges at the 10–100-ps timescale[54], and continues to grow over 1 μs[31]. Our postulate also resolves a long-standing puzzle that 2,6-difluorophenylnitrene gives similar activation energy for $^1$N → Bza as 2,3,4,5,6-pentafluorophenylnitrene, but exhibits a much lower C–H insertion yield[33]. Clearly, further pump–probe investigations are required.

## Crosslinking efficiency in model conjugated polymers
The $\xi_{XL,P}$ values can be significantly increased in conjugated polymer matrices. For example, PM6 and PBDB-T, which have practically identical chemical structures (Fig. 5a), as benzo-[1,2-b:4,5-b']dithiophene−*alt*−benzo-[1,2-c:4,5-c']dithiophene-4,8-dione (BDT−BDD) donor−acceptor polymers that are widely used in organic solar cells[55,56], give

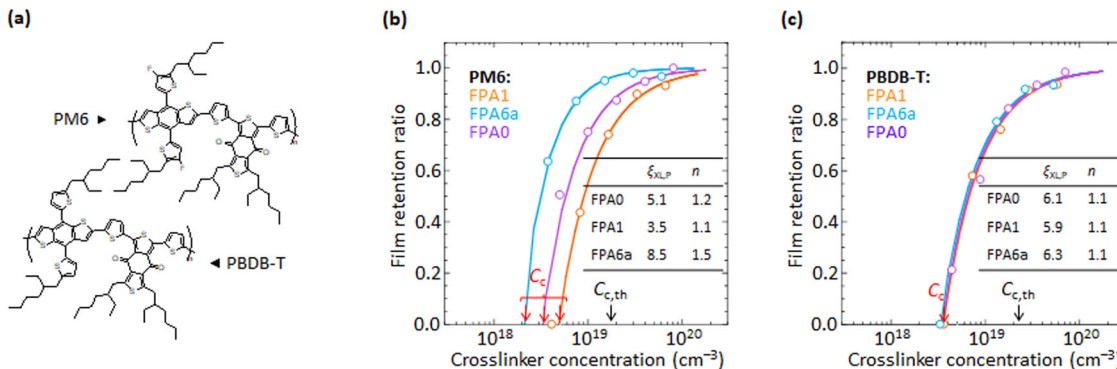

**Fig. 5 | Photocrosslinking efficiency of π-stacked polymer semiconductor models. a** Chemical structures of PM6 and PBDB-T. Film retention characteristics of **b** PM6 ($M_n$ 43 kD, Đ 2.3), and **c** PBDB-T ($M_n$ 34 kD, Đ 2.1), with selected FPAs. Data are fitted to $\zeta = 1 - (C_c/C)^n$, following Fig. 2. The theoretical gel point ($C_{c,th}$) for these PM6 and PBDB-T samples are 1.7 and $2.1 \times 10^{19}$ cm$^{-3}$, respectively, based on a polymer density of 1.2 g cm$^{-3}$.

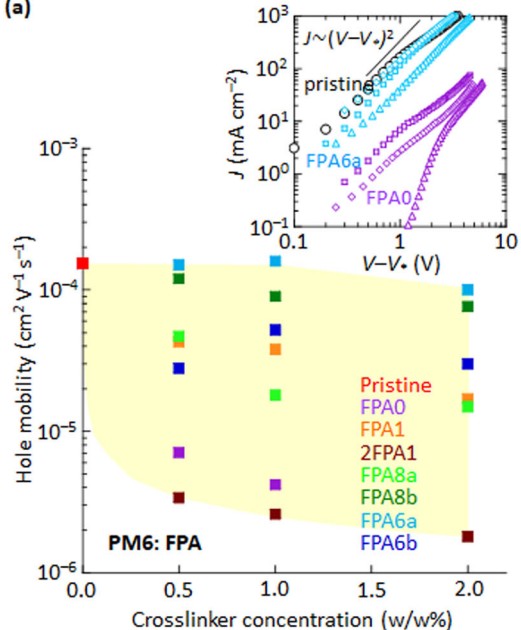

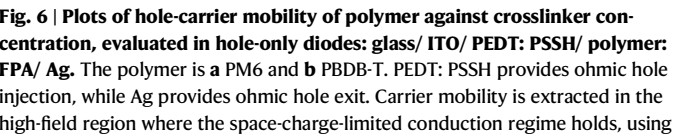

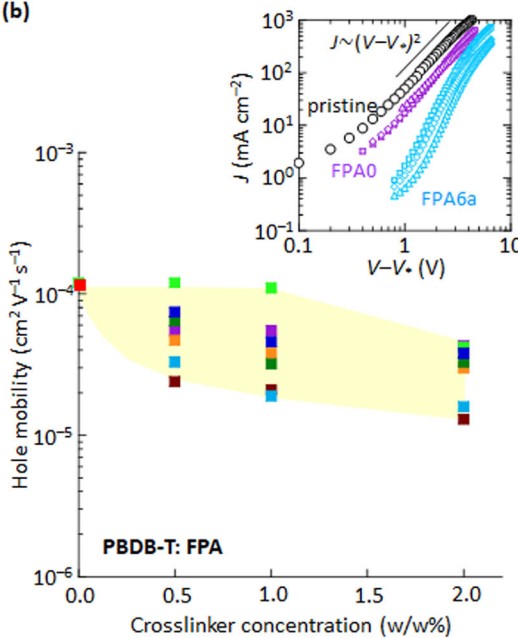

**Fig. 6 | Plots of hole-carrier mobility of polymer against crosslinker concentration, evaluated in hole-only diodes: glass/ ITO/ PEDT: PSSH/ polymer: FPA/ Ag.** The polymer is **a** PM6 and **b** PBDB-T. PEDT: PSSH provides ohmic hole injection, while Ag provides ohmic hole exit. Carrier mobility is extracted in the high-field region where the space-charge-limited conduction regime holds, using the Mott−Gurney equation: $J = \frac{9}{8}\varepsilon\mu\frac{(V-V_*)^2}{d^3}$, where $\varepsilon$ is permittivity, $d$ is thickness, $V_*$ is the apparent built-in potential, and $\mu$ is mobility, averaged over three representative diodes. Insets: Mott−Gurney plots for selected FPA crosslinkers. Legend: open squares, 0.5; diamonds, 1; and triangles, 2 w/w% of crosslinker. The yellow region gives a span of mobility quenching.

different $C_c$ behavior for the same set of photocrosslinkers (Fig. 5b, c). The two polymers share the same polymer backbone and side-chain, and thus similar π−π* band gaps (1.85 eV) and ionization energies (PM6: 5.15 eV; PBDB-T: 5.05 eV). But the fluorine substitution on the pendant thiophenes of PM6 induces better π-stacking order, as evidenced by stronger Bragg reflections[57] and higher intensity of the 620-nm π−π* band (Supplementary Fig. 2). Hence, they form a model pair that is uniquely suited to study π-stacking order effects, in particular, after selection for similar number average molar mass $M_n$ and dispersity Đ (PM6: $M_n$ 43 kDa, Đ 2.3; PBDB-T: $M_n$ 34 kDa, Đ 2.1)

All $C_c$ values are far smaller than theoretical ones. Thus, $\xi_{XL,P}$ values greatly exceed unity for these semiconducting polymers− $\xi_{XL,PBDB-T}$ are *ca.* 6.0, while $\xi_{XL,PM6}$ vary between 3.5 and 8.5−which advantageously reduces the amount of crosslinker required, minimizing adverse effects. These results are not affected by solution ageing, nor small amounts of high-boiling-solvent additives, such as 1-chloronaphthalene (Supplementary Fig. 3). Since PS-standardized gel permeation chromatography over-estimates the $M_n$ of rigid-rod

conjugated polymers (by up to tens of %)[58], $C_{c,th}$ would be underestimated, yet the $C_c$ values are smaller still by a factor of almost ten. Thus, the conclusion is that $\xi_{XL,P} \gg 1$ is robust. Literature $\zeta-C$ plots for other conjugated polymers also suggest similar findings[9]. We attribute this apparent super-efficient crosslinking to the enhancement of the effective molecular weight of the polymer by π-stacking, which protects the chains from dissolution. For comparison, the $\xi_{XL,P}$ values in PS, polyelectrolytes, and charge-doped polymers are always ≤1[59–61]. Then the $C_c$ dependence in PM6 suggests a π−π interaction between FPA moieties and the polymer backbone.

## Charge-carrier mobility quenching

Plots of the hole-carrier mobility $\mu$ of the polymer against crosslinker concentration reveal that PM6: FPA shows stronger quenching and a different quenching sequence than PBDB-T: FPA (Fig. 6). While the mobilities in the pristine film $\mu_o$ are nearly identical (*ca.* $1.5 \times 10^{-4}$ cm$^2$ V$^{-1}$ s$^{-1}$), they can fall by up to two

decades in PM6 over the crosslinkers and concentrations investigated, but only one decade in PBDB-T. Also, in a dramatic reversal of the quenching sequence, FPA6a is the weakest quencher in PM6 but the strongest in PBDB-T. Likewise, FPA8a is the weakest quencher in PBDB-T, but a moderately strong quencher in PM6.

Current–voltage ($JV$) characteristics were measured in hole-only diodes: glass/ITO/PEDT: PSSH/ polymer: FPA/ Ag, and the hole $\mu$ extracted from the space-charge-limited conduction regime, where the $\frac{\log(V - V_*)}{\log(J)}$ slope is 2.0 in the Mott–Gurney plot, with $V_*$ as the apparent built-in potential[62,63]. The pristine polymer films give similar $JV$ characteristics before and after DUV exposure, and hence similar $\mu$, indicating the absence of DUV damage (Supplementary Fig. 4). But $JV$ characteristics decrease systematically with crosslinker concentration in the crosslinked films (Supplementary Fig. 5). This is not due to chemical but morphological defects. $JV$ increases marginally after DUV photolysis of the bisazides, and decreases strongly in the presence of FPA$n$-NH$_2$ added as a crosslink simulant (Supplementary Fig. 6). Thus, hole trapping arises from the interaction between the FPA moiety/ crosslink and the polymer backbone.

## Molecular partitioning: binding affinities

We modeled the distribution as a competition between the polymer backbone and the alkyl side-chain for the FPA moiety. This competition is governed by the relative binding affinities. X-ray scattering has indeed found that FPA molecules (at concentrations above 5 w/w%) can intercalate into and disorder the π-stacks of the host polymer, for example, in P3HT: FPA1, but no change can be detected at the practical concentrations of 0.5–2 w/w%[13]. No change was observed in the π−π* absorption spectra of the films either (Supplementary Fig. 2c, d), although such concentrations do reduce the viscosity of the polymer solutions, similar to the addition of 1-chloronaphthalene.

Our model provides a basis to quantify morphological effects. To compute the van der Waals interactions between relevant molecular units, we employed the well-established OPLS4 force field[64,65]. To account for the influence of π-stacking order, we modeled both well- and less well-ordered π-stacks (Fig. 7a). FPA intercalation into the polymer backbone nanophase broadens the transport density-of-states, regardless of the specific mechanism (structural disordering, multipolar interactions or polarization effects) and quench carrier mobility (Fig. 7b). Intercalation into well-ordered π-stacks would

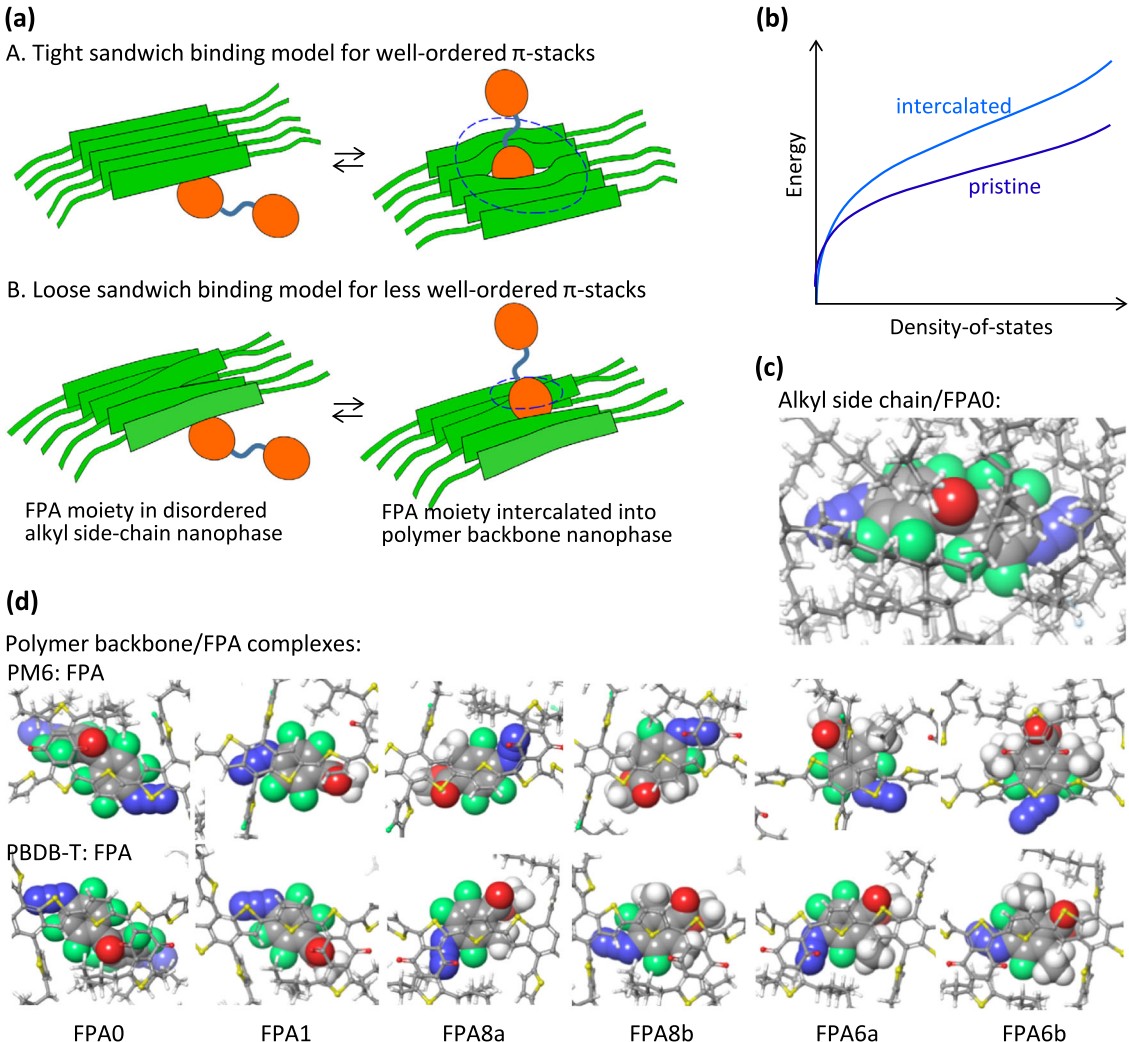

**Fig. 7 | Binding affinity model to interpret FPA molecular segregation tendencies. a** Schematic cartoon for (top) tight sandwich binding, and (bottom) loose sandwich binding. Green ribbons represent the polymer backbone; linked orange disks represent the crosslinker. The dashed circle encloses a perturbed region. **b** Schematic cartoon for intercalation on transport density-of-states. Illustrative OPLS4 molecular models for **c** FPA0 in amorphous alkyl side-chain nanophase modeled as alkane cluster and **d** polymer backbone/FPA complexes. FPA is rendered in space-filling representation for clarity. For FPA0, the entire crosslinker is computed as a unit; for the other FPAs, the hemi-crosslinker is computed, since each moiety can quasi-independently bind to the polymer backbone or side-chain phase.

produce a tight sandwich structure that incurs a loss of π...π interaction energy between the polymer segments, but intercalation into less well-ordered π-stacks would produce a loose sandwich structure without this penalty.

OPLS4 confirms FPA binds to the alkyl side-chain nanophase, and also to the polymer backbone over its BDT, BDD, and especially thiophene units, where the $C(O_2)$-ring-N axis of the FPA moiety orients at an angle to the long axis of the polymer to accommodate the C=O and any alkyl substituent (Fig. 7b). To efficiently explore the vast configuration space, we implemented a divide and conquer strategy. This strategy involves calculating the binding internal energy ($\Delta U$) of the FPA unit in small model systems that represent the key interactions, for example, in the alkyl side-chain phase, to a single polymer backbone—which corresponds to the binding energy in the loose sandwich structure, and between two polymer backbones—which corresponds to the tight sandwich structure (Supplementary Table 3). We then estimated the differential binding energy ($\Delta\Delta U$) of the FPA unit for the backbone nanophase, with the different π-stacking orders, relative to the side-chain nanophase. This approach treats realistic molecular interactions beyond the usual Flory–Huggins interaction parameters.

The subtle balance of van der Waals interactions produces nontrivial results. For the tight sandwich, the computed $\Delta\Delta U$ indicates: FPA6a, 8b > FPA6b, 8a, 1 > FPA0; while for the loose sandwich: FPA8a > FPA0, 8b > FPA1, 6a, 6b. A member to the right prefers backbone intercalation more than the one to the left. Because the entropy change ($\Delta S$) exhibits minimal variation across the FPA family, as evidenced by their nearly constant entropy of fusion ($\Delta S_{fus}$ of $1.2 \pm 0.1$ meV K$^{-1}$), the trend in the differential Gibbs free energy ($\Delta\Delta G$), which determines binding affinity, closely mirrors the $\Delta\Delta U$ trend.

### Structure–activity relationship

Figure 8 reveals a linear correlation (negative slope) between the logarithm of mobility quenching ($-\log(\mu/\mu_o)$) and the $\Delta\Delta U$ values from the tight and loose sandwich binding models for PM6 and PBDB-T, respectively. This indicates that quenching potency increases with increasing binding affinity to the backbone nanophase (more negative $\Delta\Delta U$). The linear relationships are:

$$-\log(\mu/\mu_o) = 4.45 - 7.0 * \Delta\Delta U_{PP-A}, \quad \text{for PM6 (more ordered)}$$

$$-\log(\mu/\mu_o) = 0.8 - 2.1 * \Delta\Delta U_{P-A}, \quad \text{for PBDB-T (less ordered)}$$

This finding establishes a structure–activity relationship, resolving the unexpected dependences of quenching on substitution of the FPA

guest. These models have no free parameters. Mobility quenching is decisively determined by the extent of the partitioning of the FPA moiety (guest) into the backbone nanophase. Better ordered π-stacks resist intercalation more strongly, but they exhibit a different selectivity for the guest and a stronger mobility quenching for any intercalation. As a consequence, a single universal crosslinker is not available—each polymer has its optimal crosslinker. These features reveal a fundamental nature of the morphological defect beyond the reach of mean-field theory.

Finally, the generally poor performance of the tetrafunctional 2FPA1 necessitates a different explanation. The bulky pentaerythritol core hinders the effective intercalation of 2FPA1 into polymer π-stacks. But PBDB-T: 2FPA1 exhibits a significantly lower $\mu$ after photo-exposure compared to other FPA crosslinkers (Supplementary Fig. 6d). This behavior can likely be attributed to the substantial local volume contraction upon ejection of all four $N_2$ molecules, which pulls on the π-stacks to disrupt π-stacking order and reduces carrier mobility. This suggests potential drawbacks to using tetrakis-azides and -azirines for crosslinking applications in semiconducting polymers[9,16].

### Other effects: surface enrichment and fluorescence quenching

FPAs exhibit greater surface accumulation in PM6: FPA films compared to PBDB-T: FPA films, though not enough to form a complete surface layer (Supplementary Table 4). This accords with contact angle measurements that show even 5 w/w% FPA1 in PS does not alter its surface energy[8]. Notably, FPA crosslinkers 6b and 8b effectively preserve the photoluminescence efficiency of a high-efficiency blue-emitting polymer, while 6a, 6b, 8a, and 8b achieve the same for a yellow-emitting polymer (Supplementary Fig. 7). This underscores the importance of molecular design in enabling FPA crosslinking even for sensitive light-emitting polymer layers, where maintaining photoluminescence efficiency is crucial.

### Crosslinked polymer solar cells

PM6 and PBDB-T are widely used with molecular acceptors Y6 and ITIC, respectively, for organic solar cells. We demonstrate here that the optimal FPA crosslinkers facilitate the fabrication of morphology-stabilized, acceptor-infiltrated, donor-polymer-network solar cells[2] for the high-performance PBDB-T: ITIC and PM6: Y6 photoactive systems (Fig. 9a, b). FPA0 was employed for PBDB-T and FPA6a for PM6, at 0.4 w/w%. The donor-polymer: FPA film was first deposited by spin-casting, crosslinked by DUV, developed with an appropriate solvent to remove the uncrosslinked fraction, and then back-filled with the desired acceptor by spin-casting from chlorobenzene (for ITIC) or

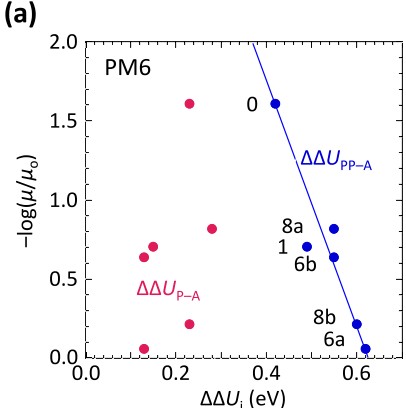

**(a)**

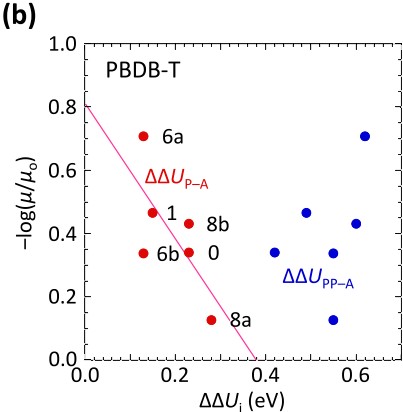

**(b)**

**Fig. 8 | Validation of structure–activity relationship.** Plots of the experimental logarithm of mobility quenching against computed differential binding energy for tight sandwich (blue) and loose sandwich (red) binding models for **a** PM6 and **b** PBDB-T. Binding energy for tight sandwich in well-ordered π-stacks ($\Delta\Delta U_{PP-A}$); and loose sandwich in less-ordered π-stacks ($\Delta\Delta U_{P-A}$), relative to amorphous alkyl side-chain nanophase (definition and data in Supplementary Table 3). $\mu$ is hole mobility in crosslinked polymer film with 1 w/w% FPA; $\mu_o$ is hole mobility in pristine film.

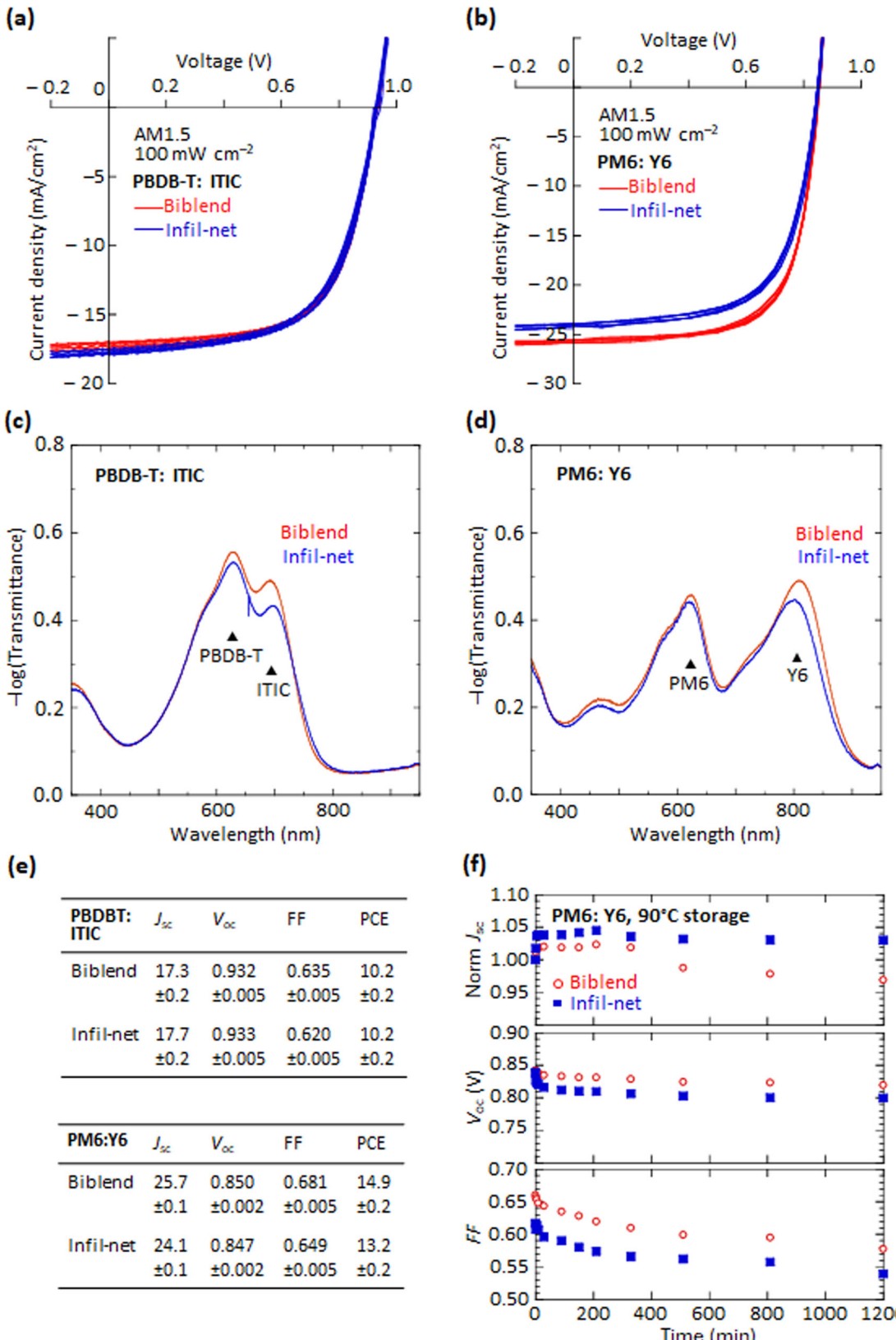

**Fig. 9 | Comparative performance of solar cells with infiltrated-polymer-network or demixed-biblend photoactive layer (PAL).** Device structure: glass/ ITO/ PEDT: PSSH/ PAL/ BT-F3NMe3: Ox, OAc/ Ag, where PAL is: **a** PBDB-T: ITIC (1:1 w/w), and **b** PM6: Y6 (1:1.2 w/w). Crosslinker concentration, 0.4 w/w%, FPA0 for PBDB-T and FPA6a for PM6. *JV* curves are shown for four typical devices. UV−Vis−NIR spectra for the PAL films: **c** PBDB-T: ITIC, and **d** PM6: Y6. **e** Statistics summary of devices where "±" denotes one standard error of the mean. $J_{sc}$, short-circuit current density (mA cm$^{-2}$); $V_{oc}$, open-circuit voltage (V); FF, fill factor; PCE, power conversion efficiency (%). **f** Thermal stability of infiltrated-network vs demixed-biblend PM6: Y6 cells. Data averaged over four diodes. The standard error is 0.5% for normalized $J_{sc}$, 0.002 V for $V_{oc}$, and 0.005 for FF.

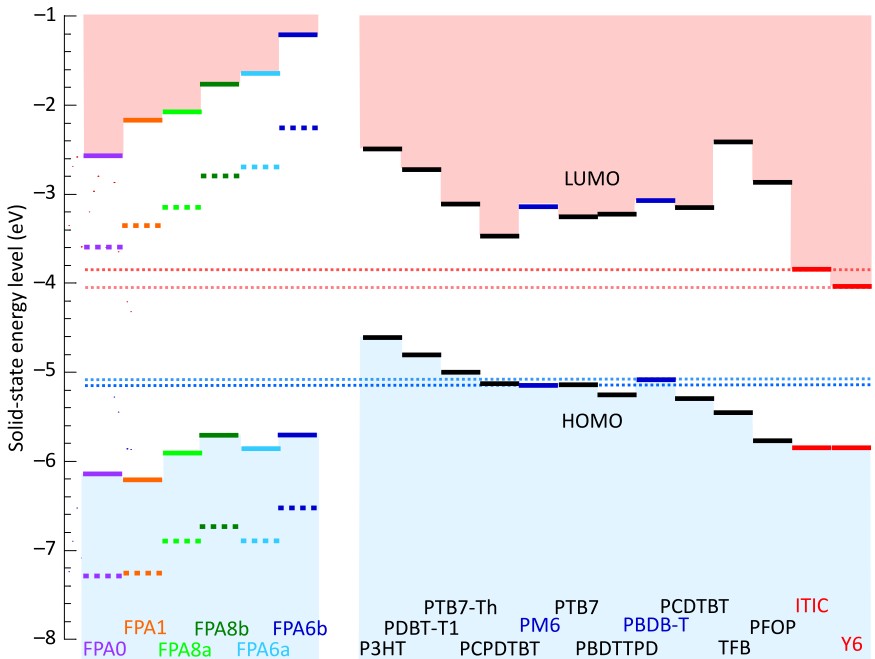

**Fig. 10 | Energy levels of FPAs and organic semiconductors.** (left) Theoretical HOMO and LUMO levels of crosslinkers: FPA moiety (dotted line) and ideal cross-link (solid line), computed in gas phase by DFT/CAM-B3LYP/6-31 G, and corrected to condensed phase by 1.5-eV inward polarization shift, i.e., HOMO(s) = HOMO(g) + 1.5 eV, and LUMO(s) = LUMO(g) − 1.5 eV, with estimated uncertainty of 0.1 eV. (right) Measured HOMO and LUMO band edges of organic semiconductors, obtained by ultraviolet photoelectron spectroscopy for HOMO band edge and estimated from bandgap and exciton binding energy correction for LUMO band edge.

chloroform (Y6) solution. A similar approach was employed recently to generate a thin top photoactive layer (NT812: Y6) for narrow-band photodiodes[12]. The film microstructure comprises an ultrafine bicontinuous network of the donor and acceptor materials that extends across macroscopic distances. Its morphology is characterized by local molecular segregation and order that is crucial for efficient exciton dissociation and charge transport, as confirmed by detailed carrier-mobility measurements (both electron and hole) for varying compositions (unpublished). The high degree of order within the acceptor phase is evidenced by the red-shifted π–π* absorption bands in both ITIC and Y6, closely resembling those of demixed-biblend films (Fig. 9c, d). For reference, control devices were fabricated in the same run by spin-casting PBDB-T: ITIC from chlorobenzene and PM6: Y6 from chloroform[66]. Despite matching efforts, the infiltrated-network films exhibited marginally lower absorbance compared to the optimized spin-cast films. PEDT: PSSH was employed as hole collection layer, and BT-F3NMe3: Ox, OAc (capped with evaporated Ag) as electron collection layer in all cells[67].

The power conversion efficiencies (PCE) of the infiltrated-network cells can approach those of the demixed-biblend cells (PBDB-T: ITIC, 10.2%; PM6: Y6, 13.2 *vs* 14.9%), comparable to those reported elsewhere[68,69], without systematic optimization. The infiltrated-network cells match the demixed-biblend cells in open-circuit voltage, and, when normalized to absorbance, also short-circuit current density, but marginally fall short in fill factor, due to a small reduction in carrier mobilities. Nevertheless, transient photocurrent measurements for the infiltrated-network PBDB-T: ITIC cells reveal a reduced population of slow carriers, suggesting a superior donor–acceptor morphology (unpublished). Clearly, infiltrated-network cells can be fabricated for non-fullerene acceptors to deliver high performance, whilst retaining the advantages of scalable fabrication and assured phase contiguity at the ultrafine length scale[2].

Crosslinking offers other potential benefits, including the fixation and stabilization of the donor–acceptor morphology, which can improve device stability[13,17,70–72]. Our infiltrated-network cells

demonstrate this concept for non-fullerene acceptors. The infiltrated-network exhibits enhanced stability in $J_{sc}$ under accelerated testing at 90 °C in the dark, likely due to morphology stabilization (Fig. 9e). Owing to the sensitivity of $J_{sc}$ to factors such as thickness and composition, normalized $J_{sc}$ is used for comparison. Nevertheless, the stability of organic solar cells is a complex, multi-faceted phenomenon, of which morphology is but one aspect. While the infiltrated-network cells outperform demixed-biblend cells in $J_{sc}$ stability, they show a faster initial decline in $V_{oc}$. Detailed work suggests surface enrichment by Y6, highlighting the need for surface engineering strategies for optimal performance (unpublished).

Finally, the HOMO level of ideal crosslinks (−N(H)-FPA-bridge-FPA-N(H)−) lies significantly deeper than the HOMO band edge of most polymer semiconductors, besides PBDB-T and PM6 (Fig. 10; see Supplementary Fig. 8 for chemical structures). Similarly, the LUMO level of the crosslinks is shallower than the LUMO band edge of numerous organic semiconductors, besides ITIC and Y6. This suggests that FPA crosslinks should not impede charge transport by trapping holes or electrons, or forming charge-transfer complexes[73,74]. While readily introduced as molecular additives for rapid screening, FPA photocrosslinkers can also be covalently attached to polymer side chains. This approach should offer additional benefits, such as higher enrichment in the alkyl side-chain nanophase. The insights gained here regarding binding affinity and molecular partitioning should remain relevant to both strategies. However, due to the fine interplay between binding in the backbone and side-chain nanophases, a library of crosslinkers with tailored properties will likely remain necessary to achieve optimal performance in different conjugated polymers and processing conditions.

## Methods
### Materials
The FPA photocrosslinkers were synthesized in-house. (Supplementary Methods). Monodispersed polystyrene standard ($M_n$ 200 kD, Đ 1.01; Sigma-Aldrich), PM6 ($M_n$ 43 kD, Đ 2.3; 1-Material), and PBDB-T ($M_n$ 34 kD, Đ 2.1; Ossila) were used as received. The light-emitting

polymers are gifts from Cambridge Display Technology Ltd. Commercial PEDT:PSSH solution (Heraeus P VP AI4083, 1:6 wt/wt) was purified before use by ion-exchange dialysis with 0.1 M hydrochloric acid, then Millipore water to remove additives and degradation products[75]. Poly((9,9-bis(3-trimethylammoniopropyl)fluorenyl-2,7-diyl)−*alt*−(benzo[2,1,3]thiadiazole-4,8-diyl)), counterbalanced with a mixture of oxalate (Ox) and acetate (OAc) at an Ox: OAc ratio of 4: 1 was synthesized in-house (BT-F3NMe3: Ox,OAc), following ref. 67. **Characterizations**. Thermogravimetry was performed on the Discovery TGA system (TA Instruments). Differential scanning calorimetry was performed on the DSC1/400 W STARe system (Mettler Toledo). FTIR spectra were collected on a nitrogen-purged Nicolet 8700 FTIR spectrometer operated in the temperature-stabilized cleanroom. **Film retention plots**. PS: FPA and other polymers: FPA films were photo-exposed to attain 99 + % photolysis. The film retention ratio was measured as the fraction of film thickness retained after solvent wash.

## Devices: general
Glass substrates with lithographically-patterned ITO were washed with semiconductor-grade acetone, then isopropanol, blown-dry with $N_2$, and cleaned by oxygen plasma (200 W, 10 min) at the point-of-use. **Hole-only diodes**. 25-nm-thick films of PEDT: PSSH were spin-cast onto clean ITO−glass substrates and baked on a digital hotplate at 140 °C for 10 min in a $N_2$ glovebox. A solution of PBDB-T in chlorobenzene (20 mg mL$^{-1}$) was prepared, heated to 80 °C for 30 min, and cooled to room temperature for 30 min at point-of-use. A solution of PM6 in chloroform (8 mg mL$^{-1}$) was prepared without heating. Appropriate volumes of crosslinker solution in chloroform (for PM6) or chlorobenzene (for PBDB-T) were mixed with the polymer solutions to get the desired crosslinker-to-polymer mass ratio. The solutions were then spin-cast over PEDT: PSSH in the $N_2$ glovebox to give 80-nm thick films, which were photo-exposed at 254-nm wavelength from a low-pressure Hg tube lamp (740 or 1100 μW cm$^{-2}$) for 5 min in the $N_2$ glovebox. Finally, 120-nm thick Ag films were evaporated through a shadow mask at a base pressure of $10^{-7}$ Torr to give eight 4.3-mm$^2$ pixels on each substrate. Current−voltage characteristics were measured using a semiconductor parameter analyzer (Keithley S4200). **Organic solar cells**. 25-nm-thick films of PEDT: PSSH were spin-cast as before. For demixed-biblend cells, a chloroform solution of PM6 (11 mg mL$^{-1}$), and of Y6 (11 mg mL$^{-1}$) containing a small amount of 1-chloronaphthalene (0.5 vol%), were mixed together to give a solution containing 1: 1.2 w/w PM6: Y6. Chlorobenzene solutions of PBDB-T and ITIC (each 20 mg mL$^{-1}$) were heated to 80 °C for 30 min, cooled to room temperature for 30 min, and then mixed together to give a solution containing 1: 1 w/w PBDB-T: ITIC. The biblend solutions were then spin-cast over the PEDT: PSSH films to give 90-nm-thick photoactive layers. For infiltrated-network cells, a chloroform solution of PM6 (7.0 mg mL$^{-1}$) containing 0.4 w/w% of FPA6a, based on the polymer, and a chlorobenzene solution of PBDB-T (20 mg mL$^{-1}$) containing 0.4 w/w% of FPA0, also based on the polymer, were prepared. These solutions were spin-cast over the PEDT: PSSH films. The polymer films were photo-exposed as before, and developed to remove the uncrosslinked materials by flood−spin-off with the appropriate solvent (chlorobenzene for PBDB-T, chloroform for PM6). A chlorobenzene solution of ITIC (13 mg mL$^{-1}$) and chloroform solution of Y6 (5.0 mg mL$^{-1}$) were prepared and spin-cast onto the polymer films at 2,000 revolutions per unit (rpm) to give the acceptor-infiltrated photoactive layers. The final thickness of the photoactive layers was about 90 nm. A 2,2,2-trifluoroethanol solution of BT-F3NMe3: Ox,OAc (2.0 mg mL$^{-1}$) was spin-cast over the photoactive layers to give a 17-nm thick layer. 120-nm thick Ag films were evaporated through a shadow mask at a base pressure of $10^{-7}$ Torr to give eight 4.3-mm$^2$ pixels on each substrate as before. *JV* characteristics of the cells were measured in a $N_2$ chamber using

a Keithley 2460 source-measure unit in the dark and under illumination by a solar simulator (Newport Sol2A). The results were spectral mismatch corrected.

## Calculations
DFT calculations were carried out using the Gaussian 09 program package. The geometry optimizations and energy calculations of perfluorophenyl azide structures and derivatives were computed at the Kohn-Sham DFT/CAM-B3LYP/6-31 G(*d*) level. For closed-shell singlet nitrene ($^1A_1$), geometry optimizations were obtained from spin-restricted DFT/CAM-B3LYP/6-31 G(*d*) wavefunctions. For open-shell singlet nitrene ($^1A_2$), initial guess orbitals for HOMO and LUMO were mixed to produce spin-unrestricted DFT/CAM-B3LYP/6-31 G(*d*) wavefunctions for geometry optimizations. For open-shell triplet nitrene ($^3A_2$), geometry optimizations were obtained from spin-unrestricted DFT/CAM-B3LYP/6-31 G(*d*) wavefunctions. The singlet-triplet splitting energies of nitrenes were finally computed by comparing zero-point energies obtained from frequency calculations of optimized singlet and triplet nitrenes. Electronic transition energies of azides and nitrenes were computed by time-dependent (TD) DFT at the CAM-B3LYP/6-31 G level and using the SMD solvation model in acetonitrile. OPLS4 calculations were performed in MacroModel (Schrödinger LLC). Initial seed configurations were generated haphazardly, held isothermally at 200 K and equilibrated for at least 1 ns, with a background dielectric constant of 1.00, after which multiple snapshots 1 ns apart were taken from the production run, and geometry optimized, to build statistics. The azide N−N−C−C dihedral angle of FPA is constrained to 10°, following DFT results.

## Reporting summary
Further information on research design is available in the Nature Portfolio Reporting Summary linked to this article.

## Data availability
Source data for all figures are available from the corresponding author.

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

## Acknowledgements

This research is supported by the National Research Foundation, Prime Minister's Office, Singapore, under its Competitive Research Program (CRP Award No. NRF-CRP24-2020-0006: A-0008375-00-00, A-0008375-01-00).

## Author contributions

L.-L.C. directed the research. R.-Q.P. supervised device fabrications. Z.-S.T. performed DFT calculations. Z.J. and H.-C.K. synthesized the crosslinkers. Z.-S.T., H.-C.K., and D.W.Y.T. characterized the crosslinkers (TG, DSC, XPS, UV–Vis–NIR, FTIR, gel dose, hole-only diodes). Z.-L.S. and H.-Y.P. fabricated and characterized the hole-only diodes and solar cells. R.-Q.P. performed the OPLS4 calculations. P.K.H.H. created the models. All authors contributed to the drafts. P.K.H.H., R.-Q.P., and L.-L.C. wrote the final manuscript.

## Competing interests

The authors declare no competing interests.
