## [Peer Review File · Nature Communications]

Optimization of fluorinated phenyl azides as universal photocrosslinkers for semiconducting polymersREVIEWER COMMENTS

Reviewer #1 (Remarks to the Author):

In this manuscript, Tan et al. investigated crosslinking efficiency of fluorinated phenyl azide (FPA) crosslinkers and their ability to crosslink π -conjugated polymers with or without harming charge carrier mobility. Interestingly, they found the unified quantum efficiency of nitrene photogeneration and photocrosslinking in FPA crosslinkers. In addition, they also demonstrated that molecular design can suppress quenching of hole carrier mobility by reducing binding affinity to polymer backbone. These obtained results confirm FPAs acting as a universal class of photocrosslinkers for polymeric electronic materials, which are timely and of interest in the field of OPV. Overall, this work provides some new information for organic photovoltaic materials and devices. Hence, I would recommend its publication in Nature Communications after fully addressing the following aspects.

1. Worth mentioning works conducted on other photocrosslinkers in OPVs.
2. The arrangement of panels in Figure 1, 2, 3, 7 is very misleading.
3. The device stability measurements involving FPA photocrosslinkers are recommended to provide in the revised manuscript.
4. The molecular formulae in Figure 1a should be more normalized and supplemented with missing molecular structures of the ITIC and Y6 materials.
5. In Figure 2, the authors attributed signals below 1200 cm^{-1} to matrix silica, but in fact, those characteristic signals of each crosslinkers below 1200 cm^{-1} were inconsistent. Could FTIR of silica be used as a reference to explain the source of characteristic peaks within this range?
6. Compared with other crosslinkers in Figure 2, FPA0-35 only possesses active content under two exposure times. Is the result reliable?
7. On the performance of the specific devices in Figure 9, we noted that the PM6 Y6 system has a significant variation in VOC. The authors should give a more detailed discussion of the variation of this parameter.
8. The effect of crosslinkers on polymeric materials is multifaceted. The authors should characterize the film microstructure in more detail, such as GIWAXS tests.
9. The authors found weaker absorption in the infile-net device than that of PDBDB-T:ITIC blend system (Figure 9), which generally induces the lower JSC. However, the JSC is increased in blend devices in this manuscript. The author should give more discussion on this point.

Reviewer #2 (Remarks to the Author):

Crosslinking of organic semiconductors is used for device patterning, imparting solvent resistance, or improving the stability of materials and devices. Photo-crosslinking can lead to changes in the chemical bonding of organic molecules due to the use of high-energy ultraviolet light, so crosslinking by heat is generally preferred. In addition, chemical crosslinking using light and heat has the possibility of deteriorating important properties of organic semiconductors such as mobility. Photo-crosslinking using FPA still has a negative effect by light irradiation, but has relatively little effect on the electro-optical properties of the crosslinked thin film, and is widely used among various photo-crosslinking systems.

In this study, the structure of FPA and its binding affinity to the polymer backbone were analyzed to investigate the effect of FPA on photo-crosslinking efficiency and mobility of organic semiconductors. It was shown that the van der Waals interaction could be controlled through the introduction of substituents into the crosslinking agent and the position change of the substituents, and through this, the quenching of hole carrier mobility can be suppressed.

It is thought that further investigation and research on the photoreaction products is necessary, but in this experiment, the photoreaction products for the compounds used was not directly investigated, and it relied on previous studies, so the experimental results are based on many assumptions. In addition, the fact that the results of this study will not have a significant impact on the options of users who actually use FPA is a factor limiting the impact of this study.

Nevertheless, it is considered to be of great significance to suggest a method to improve the crosslinking efficiency of the FPA crosslinking group and suppress the decrease in mobility through various experiments and analyzes. Accordingly, I believe that this paper deserves to be published in Nature Communications. Following issues are worth considering before publishing this work.

1. For the practical application of FPA, it is necessary to investigate the effect of the intensity and dose of light used on the organic thin film when light is irradiated on the active layer of the device.
2. It is necessary to study the effect on the lifetime of devices using cross-linked thin films.

Reviewer #3 (Remarks to the Author):

The manuscript by Tan and coauthors explores the crosslinking reactions of photocrosslinkers based on fluorinated phenyl azide (FPA) moieties, which have found widespread application in photopatterning and stabilizing various polymer semiconductors for electronic devices. Contrary to the prevailing belief that nitrene-based FPA crosslinkers are inferior to carbene-based crosslinkers in terms of crosslinking efficiency and resulting charge transport properties for conjugated polymers, this report systematically demonstrates that FPA-based crosslinkers can exhibit ideal crosslinking behavior. The study investigates a series of carefully designed FPA-based crosslinkers, aiming to elucidate the influence of the conjugating bridge group, the type and number of substituent groups on the FPA moiety, and the number of FPA moieties within the crosslinker molecules. The results presented clearly demonstrate that, with optimal design, both the photolysis efficiency of azide and the crosslinking efficiency of the crosslinker can reach unity, which was not previously expected for FPA-based crosslinkers. Additionally, the findings suggest that the binding interaction between the crosslinker and the amorphous domains of the conjugated polymer film, primarily composed of side chains, plays a role in quenching the charge carrier mobility. Careful design of the nanophase interaction within the amorphous domain can mitigate degradation in the charge carrier mobility. The manuscript is well-written, and the explanations provided are instructive, aiding readers' comprehension of the photophysical and chemical processes involving FPA-based crosslinkers. I recommend its publication in Nature Communications after addressing the following issue.

The authors investigated the influence of crosslinker structure on transport using a set of polymer semiconductors in various testbeds such as diodes (hole-only devices), transistors, and solar cells. It would be helpful if the authors could comment on their influence on the photoluminescence properties of crosslinked polymer semiconductors. I recall that paper from the same research group in 2010 (Nature Materials) that also reported the applicability of FPA-based crosslinkers to super yellow light-emitting polymers. I wonder if any valuable conclusions can be drawn regarding the luminescence characteristics of polymers based on the design of FPA-based crosslinkers. While I understand that this could open an entirely new chapter, I believe it would provide important perspectives on the works involving light-emitting polymers (e.g., Ref. 23).

On page 8, line 156, it is stated, "we found photolysis efficiency values to be unity within experimental uncertainty for all FPAs." I am curious to know how the authors determined that the photolysis efficiency value is close to unity. Is this derived from Figure 2a (the surviving azide fraction data). I believe this finding is significant, particularly in contrast to previous observations where photolysis efficiency was reported to be slight over 50%. It would be beneficial to clarify how this determination was made.

Based on the orbital energy calculation, is there a possibility for the FPAs to form nitrene upon 365 nm irradiation, despite the negligible molar absorption at that wavelength (as shown in Figure 2f)? This consideration may be valuable, especially since the 365 nm i-line irradiation set is commonly used in display industry.

Minor comment:

The dose of UV irradiation should be included (J/cm^2). The x-axis of Figure 2b should be changed to units of J/cm^2 instead of time.

It is also suggested to include recent references to the utilization of FPA-based crosslinkers for crosslinking/photopatterning of light-emitting quantum dots and perovskite nanocrystals by forming chemical bonds with the alkyl ligands on their surfaces. These works should be mentioned in the introduction.

Summary of Reviewer Comments

Reviewer #1 (Remarks to the Author):

In this manuscript, Tan et al. investigated crosslinking efficiency of fluorinated phenyl azide (FPA) crosslinkers and their ability to crosslink π -conjugated polymers with or without harming charge carrier mobility. Interestingly, they found the unified quantum efficiency of nitrene photogeneration and photocrosslinking in FPA crosslinkers. In addition, they also demonstrated that molecular design can suppress quenching of hole carrier mobility by reducing binding affinity to polymer backbone. These obtained results confirm FPAs acting as a universal class of photocrosslinkers for polymeric electronic materials, which are timely and of interest in the field of OPV. Overall, this work provides some new information for organic photovoltaic materials and devices. Hence, I would recommend its publication in Nature Communications after fully addressing the following aspects.

>> We thank the referee for support!

1. Worth mentioning works conducted on other photocrosslinkers in OPVs.

>> Fixed. Originally, we intended to sample only a small subset of the literature. Motivated by referee's comments, we have now expanded this list to direct more attention to the situation in polymer organic solar cells: McCulloch and co-workers, *Adv. Energy Mater.* (2015) on using bis-alkyl azide (DAZH) photo-crosslinker for SiIDT-BT:PCBM; Wantz and co-workers, *Adv. Mater.* (2014), on using bis-benzyl azide (BABP) also for P3HT:PCBM; Tajima and co-workers, *ACS Appl. Energy Mater.* (2023) using bis-phenyl diazirine & bis-FPA, with C10 aliphatic linker as photocrosslinkers for PM6:Y6; Ding et al, *J. Mater. Chem. A* (2013), using OBOCO – Diels-Alder thermal crosslinkers for P3HT:PCBM. We have now also included non-additive photocrosslinking in OPVs: *Sol. En. Mater. Sol. Cells* (2017) for use of terminal epoxy moieties; *Macromolecules* (2009) and *ACS Omega* (2017) for use of terminal azides; and *ACS Appl. Mater. Interf.* (2018) for use of terminal vinyls.

2. The arrangement of panels in Figure 1, 2, 3, 7 is very misleading.

>> Fixed. We have re-organized the pieces in a more logical sequence to better tell the story.

3. The device stability measurements involving FPA photocrosslinkers are recommended to provide in the revised manuscript.

>> Done. Device stability is a complex phenomenon. After one extra year looking into this problem, we appreciate that modern organic photoactive systems exhibit a number of instability mechanisms. Most of these are related to chemical instabilities of the photoactive materials and their interfaces under light, charge and thermal excitation. [Y6 is particularly problematic – there appears to be an isomerization reaction occurring in situ in the diodes.] Fixing the morphology (by crosslinking) cannot overcome these problems to make the devices generally more stable. However, we have now found evidence for the morphology fixing advantage. At 90°C, which represents accelerated testing, we found that both the blend (uncrosslinked) and infiltrated-network (crosslinked) cells evolve in the same manner for V_{oc} and FF, but the crosslinked cells show a slower rate of degradation of the J_{sc} (see Response Fig 1). As a consequence, the J_{sc} of uncrosslinked cells fall below that of the crosslinked cells after 10 h, even though they have the higher absorptance. This suggests that the degradation of J_{sc} has a morphology contribution. The data is now incorporated into the manuscript in Fig 9.

Response Fig 1. Relative thermal stability of infil-net vs biblend PM6: Y6 solar cells. Data averaged over 4 diodes. Standard error is 0.4% for normalized J_{sc} , 0.002 V for V_{oc} and 0.005 for FF.

- The molecular formulae in Figure 1a should be more normalized and supplemented with missing molecular structures of the ITIC and Y6 materials.

>> Fixed. We have added the chemical structures of ITIC and Y6, and consolidated them into Fig 9 together with those of the polymer donors. However, we have to retain the currently labeling of our FPA crosslinkers. The numbering is systematic in the time sequence of development. These same labels have been used elsewhere in publications, and work with industry. The ‘missing’ numbers are those of photocrosslinkers that did not work out.

- In Figure 2, the authors attributed signals below 1200 cm^{-1} to matrix silica, but in fact, those characteristic signals of each crosslinkers below 1200 cm^{-1} were inconsistent. Could FTIR of silica be used as a reference to explain the source of characteristic peaks within this range?

>> The board ‘bump’ feature at 1120 cm^{-1} receives a large contribution from SiO_2 in the silicon substrate. We have confirmed this from the spectrum of the substrate itself. But the SiO_2 content (intrinsic oxide and residual bulk oxygen) varies between silicon wafers. This region is overlaid with the spectra of PS and the FPA crosslinkers. The appearance varies with sample because of the difference in SiO_2 content and in crosslinker. However, the

difference spectra that we presented to track changes at partial and complete photolysis were collected on the *same* sample for different photolysis times. Thus, the changes reflect only the photolysis and crosslinking processes. We have improved the figure caption to explain this more carefully.

6. Compared with other crosslinkers in Figure 2, FPA0-35 only possesses active content under two exposure times. Is the result reliable?

>> FPA0 was the first crosslinker on which we measured the quantum yield a few years ago. The experimental conditions were not so optimal then – the film was thin, and the number of measurements small. Prompted by the referee, we have now repeated the measurement on a thicker film with more data. The old PFA0-35 dataset in Figure 2a is now replaced with the new FPA0-400 dataset. The conclusions are identical. The nitrene photogeneration efficiency is 1.00 ± 0.05 . This result is very reliable.

7. On the performance of the specific devices in Figure 9, we noted that the PM6 Y6 system has a significant variation in VOC. The authors should give a more detailed discussion of the variation of this parameter.

>> We have collected another dataset. We can now confirm that the initial difference in V_{oc} is small, less than 0.005 V. The new dataset is used in Fig 9. In separate work, we know that V_{oc} is sensitive to surface enrichment of the components. An increase in the acceptor concentration at the top surface will decrease V_{oc} by decreasing V_{bi} by causing deeper Fermi level pinning if that top surface forms the electron contact. The infiltrated-network cells tend to raise acceptor concentration at the surface relative to the spin-cast blend cells. These issues are complex, and will be discussed in detail in a separate publication.

8. The effect of crosslinkers on polymeric materials is multifaceted. The authors should characterize the film microstructure in more detail, such as GIWAXS tests.

>> This is a complex issue. Literature data suggests that the degree of crystallinity and quality of π -stacking are both not affected at the crosslinker concentrations used here (< 1 w/w%). Thus, the FPA crosslinkers do not disrupt the π -stacks. But GIWAXS is not sensitive to the paracrystalline/ amorphous domains where interactions may occur more strongly. To characterize this aspect, we have performed detailed measurements of carrier mobilities (both electron and hole) to ascertain the connectivity and molecular order for charge transport as a function of film composition of the crosslinked network. We found the crosslinked network lowers the percolation threshold for the molecular acceptor and keeps high electron mobility over a wider composition range than in the spin-cast biblend films. See Response Fig 2. The crosslinked network also maintains a high hole mobility generally above that of the neat film. See Response Fig 3. These results reveal that the film microstructure comprises intertwining phases that percolate over macroscopic distances, with local molecular segregation and order necessary for charge transport. The latter is demonstrated by the red-shifted π - π^* absorption bands for both ITIC and Y6 in the infiltrated network, similar to that of the spin-cast biblend films. We have now included more discussion in the manuscript now. Detailed analysis is complex and will be published elsewhere.

Response Fig 2. Plot of SCLC electron mobility of ITIC in PBDB-T: ITIC biblend and infiltrated-network films against acceptor volume fraction. Network crosslink density, $1 \times 10^{19} \text{ cm}^{-3}$. Device configuration: ITO/PEDT:PSSCs/PAL/EIL1/Ag. Lines are guides-to-the-eye.

Response Fig 3. Plot of SCLC hole mobility of PBDB-T in PBDB-T: ITIC biblend and infiltrated-network films against acceptor volume fraction. Network crosslink density, $1 \times 10^{19} \text{ cm}^{-3}$. Device configuration: ITO/PEDT:PSSH/PBDB-T:ITIC/Ag.

9. The authors found weaker absorption in the infil-net device than that of PDBDB-T:ITIC biblend system (Figure 9), which generally induces the lower JSC. However, the JSC is increased in biblend devices in this manuscript. The author should give more discussion on this point.

>> This is another complex issue that will be discussed more thoroughly in a dedicated paper. The J_{sc} depends on absorption quantum efficiency, photocarrier generation efficiency and photocarrier collection efficiency. In the case of PBDB-T: ITIC cells, the infiltrated network shows a slightly lower average absorption efficiency, but a slightly larger J_{sc} . This suggests that the photocarrier generation and/or collection efficiency has improved. The improvement is small, and we could not presently separate the two effects. But we have found evidence from transient photocurrent spectroscopy for reduction of the fraction of slow carriers in the infiltrated-network cells. See Response Fig 4. We have now added more discussion in the manuscript.

Response Fig 4. Transient photocurrent measurement of PBDB-T: ITIC solar cell. Normalized photocurrent decay plot of biblend (red) and infil-net (blue) PBDB-T: ITIC devices. Cell configuration: glass/ ITO/ 20-nm PEDT:PSSH/ 90-nm PBDB-T: ITIC (1:1 w/w)/ 10-nm N2/ Ag. Inset: zoom in for the 0–1 μs . Two devices are shown for each type. The time of the initial fall of photocurrent T_0 is the same for both biblend and infiltrated-network cells. The photocurrent at 20 μs is also similar. But the photocurrent decays faster for infiltrated-network cells, suggesting faster photocarrier extraction, and thus superior donor–acceptor morphology. Excitation, 654-nm LED. Pulse: square, 16- μs wide, 10-ms repeat.

Reviewer #2 (Remarks to the Author):

Crosslinking of organic semiconductors is used for device patterning, imparting solvent resistance, or improving the stability of materials and devices. Photo-crosslinking can lead to changes in the chemical bonding of organic molecules due to the use of high-energy ultraviolet light, so crosslinking by heat is generally preferred. In addition, chemical crosslinking using light and heat has the possibility of deteriorating important properties of organic semiconductors such as mobility. Photo-crosslinking using FPA still has a negative effect by light irradiation, but has relatively little effect on the electro-optical properties of the crosslinked thin film, and is widely used among various photo-crosslinking systems.

>> The referee is indeed right that photocrosslinking is generally thought to introduce chemical defects into the organic semiconductor. A key aim of this report is to document that the density of such defects can be controlled to very low levels in optimal cases, because FPA photocrosslinking has practically 100 % quantum efficiency for nitrene generation and for crosslinking. The bigger challenge is morphological defects. The other key aim of this report is to show that the density of such defects can also be reduced to acceptable levels through molecular design of the FPA crosslinker. As a consequence, photo-crosslinking can be surprisingly less harmful than thermal crosslinking for semiconductor polymers. Nevertheless, thermal crosslinking is still required for thick films. We have been running a thermal crosslinking research programme for some time now. The challenge is the limited quantum efficiencies and the relatively high temperatures (>140°C) required, which can disrupt certain morphologies, e.g. polymer donor–molecular acceptor. The insights gained here would also help the design of suitable thermal crosslinkers for polymer semiconductors.

In this study, the structure of FPA and its binding affinity to the polymer backbone were analyzed to investigate the effect of FPA on photo-crosslinking efficiency and mobility of organic semiconductors. It was shown that the van der Waals interaction could be controlled through the introduction of substituents into the crosslinking agent and the position change of the substituents, and through this, the quenching of hole carrier mobility can be suppressed.

It is thought that further investigation and research on the photoreaction products is necessary, but in this experiment, the photoreaction products for the compounds used was not directly investigated, and it relied on previous studies, so the experimental results are based on many assumptions. In addition, the fact that the results of this study will not have a significant impact on the options of users who actually use FPA is a factor limiting the impact of this study.

>> The referee is also right that it is important to study the photoreaction products. But it is very challenging to do so in the solid films with actual target polymers. This is why we have developed and implemented the concept of in-situ photoreaction product analysis using FTIR, using polystyrene as model polymer (for example, the data in Fig 2 of the report). We analyze the disappearance and appearance of sensitive IR modes to infer the site of crosslinking and nature of products formed. This has allowed us to show that singlet reaction products are absent, so are the ketenimine derivatives. Thus, the primary crosslinking sites occur in the alkyl side chains. As a consequence, we are able (finally) to explain to FPA users the key considerations and molecular design rules for optimal FPA crosslinking, which we hope would be of value to the research and technology communities.

Nevertheless, it is considered to be of great significance to suggest a method to improve the crosslinking efficiency of the FPA crosslinking group and suppress the decrease in mobility through various experiments and analyzes. Accordingly, I believe that this paper deserves to be published in Nature Communications. Following issues are worth considering before publishing this work.

>> We thank the referee for the kind words!

1. For the practical application of FPA, it is necessary to investigate the effect of the intensity and dose of light used on the organic thin film when light is irradiated on the active layer of the device.

>> Yes, we completely agree with the referee. In fact, this is part of our standard protocol when testing new materials. Over the last ten years or so, we found that almost all polymer semiconductor films can withstand the required DUV dose, provided the exposure is conducted in nitrogen to avoid photo-oxidation. For example, the diode properties of PBDB-T and PM6 when irradiated with DUV are shown in Response Fig 5. The active layer gives practically identical J-V characteristics with and without the DUV irradiation, even at an excessive dose of 200 mJ cm^{-2} . [The usual required dose is 100 mJ cm^{-2}]. This experiment is sensitive to changes in carrier mobility and injection efficiency. Any possible change in carrier mobility is evaluated to be within $\pm 30\%$, which is close to the uncertainty introduced by film thickness and processing. The data is now included as Supplementary Fig 5.

Response Fig 5. Hole-only diodes to evaluate effect of 254-nm DUV on hole-carrier mobility: glass/ITO/PEDT:PSSH/polymer/Ag. (a) PBDB-T (red, pristine; blue, after DUV) and (b) PM6 (red, pristine; blue, after DUV). Spin-on PEDT:PSSH provides ohmic hole injection contact, evaporated Ag provides hole exit contact. Films were photo-exposed at 254-nm wavelength from a low-pressure Hg lamp (dose, 200 mJ cm^{-2}) in a nitrogen glovebox before Ag deposition. Second sweep data in the high-to-low direction are shown for representative diodes. Film thicknesses, 100 nm (for PBDB-T), 75 nm (PM6). Insets: Mott-Gurney plots.

2. It is necessary to study the effect on the lifetime of devices using cross-linked thin films.

>> Done. Device stability is a complex phenomenon. After one extra year looking into this problem, we appreciate that modern organic photoactive systems exhibit a number of instability mechanisms. Most of these are related to chemical instabilities of the photoactive

materials and their interfaces under light, charge and thermal excitation. [Y6 is particularly problematic – there appears to be an isomerization reaction occurring in situ in the diodes.] Fixing the morphology (by crosslinking) cannot overcome these problems to make the devices generally more stable. However, we have now found evidence for the morphology fixing advantage. At 90°C, which represents accelerated testing, we found that both the biblend (uncrosslinked) and infiltrated-network (crosslinked) cells evolve in the same manner for V_{oc} and FF, but the crosslinked cells show a slower rate of degradation of the J_{sc} (see Response Fig 1). As a consequence, the J_{sc} of uncrosslinked cells fall below that of the crosslinked cells after 10 h, even though they have the higher absorptance. This suggests that the degradation of J_{sc} has a morphology contribution. The data is now incorporated into the manuscript in Fig 9.

Reviewer #3 (Remarks to the Author):

The manuscript by Tan and coauthors explores the crosslinking reactions of photocrosslinkers based on fluorinated phenyl azide (FPA) moieties, which have found widespread application in photopatterning and stabilizing various polymer semiconductors for electronic devices. Contrary to the prevailing belief that nitrene-based FPA crosslinkers are inferior to carbene-based crosslinkers in terms of crosslinking efficiency and resulting charge transport properties for conjugated polymers, this report systematically demonstrates that FPA-based crosslinkers can exhibit ideal crosslinking behavior. The study investigates a series of carefully designed FPA-based crosslinkers, aiming to elucidate the influence of the conjugating bridge group, the type and number of substituent groups on the FPA moiety, and the number of FPA moieties within the crosslinker molecules. The results presented clearly demonstrate that, with optimal design, both the photolysis efficiency of azide and the crosslinking efficiency of the crosslinker can reach unity, which was not previously expected for FPA-based crosslinkers. Additionally, the findings suggest that the binding interaction between the crosslinker and the amorphous domains of the conjugated polymer film, primarily composed of side chains, plays a role in quenching the charge carrier mobility. Careful design of the nanophase interaction within the amorphous domain can mitigate degradation in the charge carrier mobility. The manuscript is well-written, and the explanations provided are instructive, aiding readers' comprehension of the photophysical and chemical processes involving FPA-based crosslinkers. I recommend its publication in Nature Communications after addressing the following issue.

>> We thank the referee for the analysis and appreciation. We also thank the referee for support!

The authors investigated the influence of crosslinker structure on transport using a set of polymer semiconductors in various testbeds such as diodes (hole-only devices), transistors, and solar cells. It would be helpful if the authors could comment on their influence on the photoluminescence properties of crosslinked polymer semiconductors. I recall that paper from the same research group in 2010 (Nature Materials) that also reported the applicability of FPA-based crosslinkers to super yellow light-emitting polymers. I wonder if any valuable conclusions can be drawn regarding the luminescence characteristics of polymers based on the design of

FPA-based crosslinkers. While I understand that this could open an entirely new chapter, I believe it would provide important perspectives on the works involving light-emitting polymers (e.g., Ref. 23).

>> We thank the referee for pointing us to this important question. The entire set of crosslinkers has now been tested with proprietary commercial blue and yellow polymers developed some time ago by CDT. The films were photocrosslinked on fused silica substrates at 0.1 wt/wt% crosslinkers. A photograph was then taken of the crosslinked films side-by-side, illuminated by the low-pressure Hg lamp tube. The intensity of the photoluminescence emitted through the front face of the film provides a measure of relative photoluminescent quenching, if any. Please see Response Fig 6. FPA6b and 8b show no photoluminescence quenching at all for the blue polymer, while FPA6a, 6b, 8a and 8b show no quenching for the yellow polymer. This is a significant improvement over the earlier situation. It demonstrates that ‘benign’ photocrosslinkers are now available. The data is now included in Supplementary Fig 8. We shall be very delighted to collaborate with and develop this further if the referee is interested.

Response Fig 6. Photoluminescence image of blue and yellow light-emitting polymer semiconductors, excited at 254-nm wavelength. Films containing 0.1 wt/wt % of FPA crosslinkers were crosslinked by photo-exposure at 254-nm wavelength (dose, 200 mJ cm^{-2}) in nitrogen glovebox, and imaged at 254-nm excitation. L gives the lightness value in the hue–saturation–lightness (HSL) colour model for the front-face emission. No fluorescence quenching occurs for FPA6b and 8b with the blue polymer, and for FPA6a, 6b, 8a and 8b with the yellow polymer.

On page 8, line 156, it is stated, "we found photolysis efficiency values to be unity within experimental uncertainty for all FPAs." I am curious to know how the authors determined that the photolysis efficiency value is close to unity. Is this derived from Figure 2a (the surviving azide fraction data). I believe this finding is significant, particularly in contrast to previous observations where photolysis efficiency was reported to be slight over 50%. It would be beneficial to clarify how this determination was made.

>> The referee is right. The photolysis efficiency is evaluated from the time constant for the surviving azide fraction data according to: $\phi_P = t_{o,th} t_o^{-1}$, as explained in the footnote of Table 2, where t_o is the experimental photolysis time constant and $t_{o,th}$ is the theoretical time constant given by $t_{o,th} = (E_q \xi \sigma / 2)^{-1}$ for bisazide and $(E_q \xi \sigma / 4)^{-1}$ for tetrakisazide, where E_q is the photon flux, σ is the absorption cross section, and ξ is the absorption enhancement factor due to thin-film interference effects, and the divisors 2 or 4 account for number of azide groups. We know σ from solution-state UV absorption spectroscopy, and E_q from SiC photodiode measurement of the exposure intensity.

Based on the orbital energy calculation, is there a possibility for the FPAs to form nitrene upon 365 nm irradiation, despite the negligible molar absorption at that wavelength (as shown in Figure 2f)? This consideration may be valuable, especially since the 365 nm i-line irradiation set is commonly used in display industry.

>> The FPAs in this work, except for FPA0, have negligible absorption at 365-nm wavelength. It seems not possible use 365-nm wavelength (3.4 eV) to access the lowest-lying $^1N_3^*$ state of the azide (> 3.7 eV; Fig 3) required for N_2 ejection. However, it is possible to downshift the energy of this state to lower energies (< 3.4 eV) by conjugation extension. Indeed, we have developed red-shifted FPAs through conventional conjugation extension using α,β -unsaturated ketone linkages. A portion of this work has been published in: *J. Mater. Chem. C*, **2020**, 8, 253. These photocrosslinkers can be photo-excited with 365-nm wavelength, but their photophysical properties have not been fully characterized yet. The change in the π -conjugation system seems to promote stronger molecular interaction with those hole-transport polymers that we have

tested, leading to stronger hole-trapping and photoluminescence quenching. In principle, the insights gained from the present work can be used to design better i-line photocrosslinkers.

Minor comment:

The dose of UV irradiation should be included (J/cm^2). The x-axis of Figure 2b should be changed to units of J/cm^2 instead of time.

>> Done. We thank the referee for this suggestion. We had originally plotted against time to show how the time constant was extracted. Plotting against dose has a further advantage for comparison across labs. We have done that now.

It is also suggested to include recent references to the utilization of FPA-based crosslinkers for crosslinking/photopatterning of light-emitting quantum dots and perovskite nanocrystals by forming chemical bonds with the alkyl ligands on their surfaces. These works should be mentioned in the introduction.

>> Done. We thank the referee for pointing out this aspect.

REVIEWERS' COMMENTS

Reviewer #1 (Remarks to the Author):

The authors have addressed most of the concerns and the revised manuscript is certainly improved. It may be acceptable for publication.

Reviewer #2 (Remarks to the Author):

The previous referee's questions and comments were mostly well reflected in the revised manuscript. Therefore, I am pleased to recommend this paper for publication in Nature Communications with this form.

Reviewer #3 (Remarks to the Author):

I appreciate all the efforts made by the authors for the revision. The revised manuscript thoughtfully addressed the issues that myself and other reviewers have raised. Now, I suggest its publication to Nature Communications.

You may want to reference an article that first utilized FPA-based crosslinkers for crosslinking/photopatterning quantum dots (<https://www.nature.com/articles/s41467-020-16652-4>).